# Foveated Dynamic Transformer: Robust and Efficient Perception Inspired by the Human Visual System

## Abstract

The human visual system (HVS) employs foveated sampling and eye movements to achieve efficient perception, conserving both metabolic energy and computational resources. Drawing inspiration from this efficiency, we introduce the *Foveated Dynamic Vision Transformer (FDT)*, a novel architecture that integrates these mechanisms into a vision transformer framework. Unlike existing models, the FDT uses a single-pass strategy, utilizing fixation and foveation modules to enhance computational efficiency and accuracy. The fixation module identifies fixation points to filter out irrelevant information, while the foveation module generates foveated embeddings with multi-scale information. Our findings show that the FDT achieves superior accuracy and computational efficiency, with a 34% reduction in multiply-accumulate operations. Additionally, the FDT exhibits robustness against various types of noise and adversarial attacks without specific training for these challenges. These attributes make the FDT a significant step forward in creating artificial neural networks that mirror the efficiency, robustness, and adaptability of the HVS. [1]

## 1 Introduction

Recent studies indicate that deep neural networks and the human brain interpret the environment differently, with the human visual system (HVS) dynamically filtering task-irrelevant information to focus on potential objects of interest, a process that leads to greater robustness against imperceptible perturbations that can mislead neural networks (Dodge & Karam, 2017; Azulay & Weiss, 2019; Szegedy et al., 2014; Carlini & Wagner, 2017). The retina contains photoreceptors, with the fovea—a high spatial resolution area—playing a key role in color perception and visual detail recognition (Curcio et al., 1990). The highest photoreceptor density at the fovea decreases with eccentricity, resulting in a variable-resolution image transmitted to the brain, a phenomenon known as foveation, highlighting HVS's multi-resolution perception. Studying HVS to enhance deep neural network design is therefore a promising research avenue for developing intelligent agents.

Approximately $10^7$ to $10^8$ bits of information enter the visual nerve every second in the HVS (Itti & Koch, 2001). To manage this data efficiently, the HVS uses saccadic eye movements to direct the fovea to selected targets, creating a detailed scene map from varied resolutions, known as fixation points, and saving computational resources (Itti & Koch, 2001; Bruce & Tsotsos, 2009). Inspired by the HVS, several studies have incorporated foveation and fixation mechanisms into neural networks (Mnih et al., 2014; Akbas & Eckstein, 2017; Thavamani et al., 2021). Existing methods sequentially process the input image, first locating the fixation points and then processing the features around the fixation points. However, these approaches are not optimal for two reasons. First, several inferences are required for each fixation point. Second, they require a fusion mechanism to exploit the collective information acquired from various fixation points.

We therefore propose a biologically inspired transformer architecture dubbed Foveated Dynamic Transformers (FDT) comprised of foveation and fixation modules that dynamically select multiscale tokens based on the input image. To simulate foveation in HVS, we process input tokens with

---

[1]The code will be shared upon acceptance.

Biological Eye Movements

Artificial Eye Movements

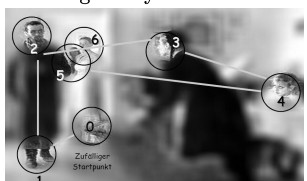 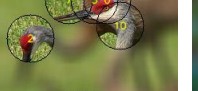 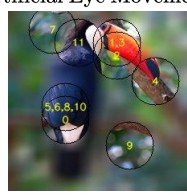 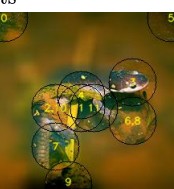

Figure 1: Illustration comparing the fixations of the human visual system (left; Yarbus et al. (1967)) and an ANN (right). Areas outside the fixations are blurred to highlight regions of interest. The sequence of eye movements is indicated by numbers. Notably, the ANN exhibits overlapping fixation points, as shown by multiple numbers at the same location, separated by commas.

the foveation module to generate multiscale queries, keys, and values. Inspired by the radial-polar pooling model of foveation proposed in (Freeman & Simoncelli, 2011), foveation module transforms the token into several scales with increasing window size. We employ dynamic networks to simulate eye movement, with the dynamic fixation module producing a fixation map for each token in each transformer block. Tokens that are not at the fixation point are discarded. The multi-head attention processes only the remaining tokens. The processed tokens that are located at the fixation points are merged with the non-fixated tokens and are sent to the next block. Multiple blocks in transformer process information from multiple fixation points by combining the information transferred from the previous blocks, which enables to implement foveation and fixation mechanisms in single pass.

Using a DeiT architecture as a baseline, we evaluate our model on an image classification task. We integrate the fixation and foveation modules into the same architecture and evaluate the effectiveness, efficiency, and robustness of our model on the ImageNet100 database. The FDT architecture enhances the robustness of the vision transformers against adversarial attacks, shortcut learning, and natural corruption by 27%, 6%, and 3%, respectively, without being directly trained for these specific challenges. Moreover, FDT achieves a 34% reduction in computational demand, measured in Multiply-Accumulate operations (MACs), demonstrating its efficiency and effectiveness in processing while maintaining a lean computational footprint.

## 2 FOVEATED DYNAMIC TRANSFORMER (FDT)

We introduce the FDT, a bio-inspired transformer with two additional modules: Foveation and Fixation. The *Foveation module* mimics the human visual system's multi-resolution perception by embedding input features at various resolutions. The *Fixation module* selectively processes tokens through gaze sampling, producing binary decisions for each foveated query. The MHSA module then processes a subset of the foveated queries, keys, and values based on the Fixation module's decisions, while the remaining tokens are passed to the next block. The FDT architecture is shown in Figure 2.

### 2.1 FOVEATION MODULE

The Foveation module in the FDT architecture is designed to replicate the foveation mechanism in the human visual system by incorporating multi-resolution information from neighboring features to generate query, key, and value features (Figure 3). This mechanism enables the perception of varying resolutions in both the central and peripheral visual fields. The Foveation module consists of multiple blockwise separable convolutional layers, each of which progressively extracts information from an expanding receptive field. In order to process auxiliary tokens, such as classification tokens, in convolutional layers, we reorganize input tokens $t$ into patch tokens in image form ($t_p$) and auxiliary tokens ($t_a$) using the function $\mathcal{K}$, as follows:

$$t_a, t_p = \mathcal{K}(t) \quad \text{where} \quad \mathcal{K} : \mathbb{R}^{C \times N} \mapsto \mathbb{R}^{C \times A}, \mathbb{R}^{C \times H \times W}. \tag{1}$$

where $C$ represents the size of the embedding, $N$ is the number of tokens, $A$ is the number of auxiliary tokens, and $H$ and $W$ are the height and width of the token.

The Foveation module leverages multi-resolution information by applying successive depthwise separable convolutions ($\mathcal{DSC}$) to patch tokens $T_p$ and pointwise convolutions ($\mathcal{PC}$) to auxiliary tokens

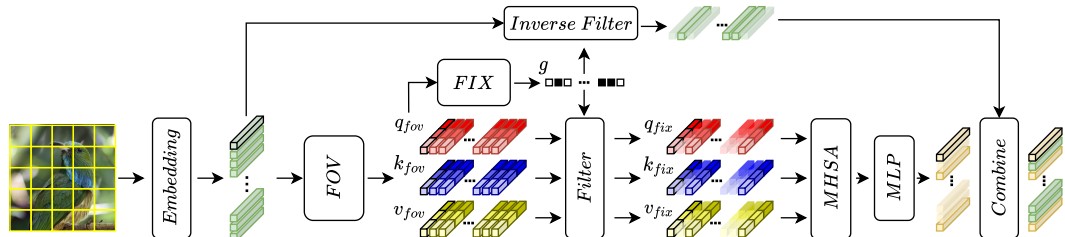

Figure 2: Schematic of the *Foveated Dynamic Transformer (FDT)*, our novel approach inspired by the HVS. FDT integrates fixation and foveation mechanisms within a single processing pass, **eliminating the need for iterative passes**. Input tokens undergo foveation via the *FOV* module, sampling features at varying resolutions and yielding foveated query, key, and value vectors. The *FIX* module then uses these vectors to identify specific fixation points for targeted token processing. The fixation map $g$ filters foveated features within the *MHSA* module. After processing through an *MLP* block, the architecture combines selectively processed fixated tokens with the remainder, positioning processed tokens at fixation points and unprocessed tokens in their original locations. This mechanism mimics human visual attention patterns, enhancing the efficiency and interpretability of neural networks.

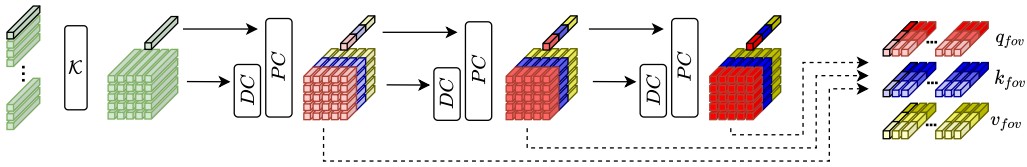

Figure 3: **Foveation module** comprises multiple blockwise separable convolutional layers with progressively larger receptive fields. It processes auxiliary and patch tokens using depthwise ($\mathcal{DC}$) separable convolutions and pointwise ($\mathcal{PC}$) convolutions. The module then splits and merges these features to form foveated-query, -key, and -value features, incorporating multi-scale information.

$T_a$, as the latter are two-dimensional features. This is mathematically represented as follows:

$$t_p^l = \mathcal{DSC}^{l-1}(t_p^{l-1}) \qquad \text{and} \qquad t_a^l = \mathcal{PC}^{l-1}(t_a^{l-1}). \tag{2}$$

Here, $l$ refers to the layer and $\mathcal{DSC}$ divides a kernel into two independent kernels that perform depthwise and pointwise convolutions, $\mathcal{DSC}(x) = \mathcal{PC}(\mathcal{DC}(x))$. As a result, each successive layer of the Foveation module employs progressively larger receptive fields.

To embed multi-scale information into the query, key, and value features, we split the auxiliary and patch-related features of each layer into three equal-sized splits in the channel dimension using function $\mathcal{S}$:

$$
\begin{aligned}
t_{aq}, t_{ak}, t_{av} &= \mathcal{S}(t_a); & \mathcal{S} &: \mathbb{R}^{C \times A} \mapsto \mathbb{R}^{C/3 \times A}, \mathbb{R}^{C/3 \times A}, \mathbb{R}^{C/3 \times A}, \\
t_{pq}, t_{pk}, t_{pv} &= \mathcal{S}(t_p); & \mathcal{S} &: \mathbb{R}^{C \times H \times W} \mapsto \mathbb{R}^{C/3 \times H \times W}, \mathbb{R}^{C/3 \times H \times W}, \mathbb{R}^{C/3 \times H \times W}.
\end{aligned}
\tag{3}
$$

Finally, we merge and concatenate the auxiliary and patch splits from all three levels in the channel dimension to form foveated-query ($q_f$), -key ($k_f$), and -value ($v_f$) features using the inverse of the operation that was applied for initial rearrangement:

$$
\begin{aligned}
q_{fov} &= [\mathcal{K}^{-1}(t_{aq}^0, t_{pq}^0) \mid \cdots \mid \mathcal{K}^{-1}(t_{aq}^{l-1}, t_{pq}^{l-1}) | \mathcal{K}^{-1}(t_{aq}^l, t_{pq}^l)], \\
k_{fov} &= [\mathcal{K}^{-1}(t_{ak}^0, t_{pk}^0) \mid \cdots \mid \mathcal{K}^{-1}(t_{ak}^{l-1}, t_{pk}^{l-1}) | \mathcal{K}^{-1}(t_{ak}^l, t_{pk}^l)], \\
v_{fov} &= [\mathcal{K}^{-1}(t_{av}^0, t_{pv}^0) \mid \cdots \mid \mathcal{K}^{-1}(t_{av}^{l-1}, t_{pv}^{l-1}) | \mathcal{K}^{-1}(t_{av}^l, t_{pv}^l)].
\end{aligned}
\tag{4}
$$

## 2.2 FIXATION MODULE

Humans execute a sequence of eye movements to construct a detailed scene map, selecting fixation points based on multi-resolution information from foveated perception. Inspired by this, we introduce

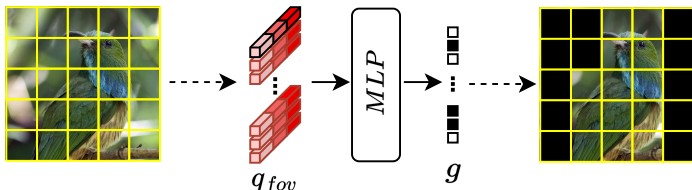

Figure 4: **Fixation module** utilizes multi-resolution information from the foveated-query token $q_{fov}$ through a single linear layer, generating logits for the binary decision of identifying fixation points. The resulting fixation map then filters the foveated features for processing within the MHSA module.

a fixation module that leverages this multi-resolution data to determine the processing input token, mirroring the human brain's decision-making.

Our fixation module uses the multi-resolution information in the foveated-query token $t$ to identify fixation points. This token's extensive receptive field allows for fixation decisions to be based on a single token. We implemented the module with a simple neural network featuring a single linear layer, which processes the foveated query and outputs two logits for binary fixation decisions.

$$\ell_{fix}^{i,j} = \text{MLP}(q_{fov}^{i,j}) \quad \forall i \in \{0, ..., H-1\}, \forall j \in \{0, ..., W-1\}. \tag{5}$$

To generate a fixation map, we feed all the foveated query features to the fixation module (FIX). The fixation map is generated based on the logits produced by the module. The position where the first logit value is higher than the second logit value is set to 1.

$$g(i,j) = \begin{cases} 1, & \ell_{fix}^{i,j}(0) > \ell_{fix}^{i,j}(1), \\ 0, & otherwise. \end{cases} \tag{6}$$

The fixation map is used to filter the foveated features that are processed in the Multi-Head Self-Attention (MHSA) module. See Figure 4.

## 2.3 OVERALL ARCHITECTURE

HVS uses foveation and fixation mechanisms to efficiently process information. Inspired by this, we developed the Foveated Dynamic Transformer (FDT), a variant of the standard vision transformer that integrates these models into its architecture. FDT retains the use of MHSA and MLP modules but processes only fixated tokens. Data flow in FDT, denoted by layer norm (LN), is structured as:

$$q_{fov}, k_{fov}, v_{fov} = \text{FOV}(\text{LN}(x)), \quad g = \text{FIX}(q_{fov}), \tag{7}$$

$$q_{fix}, k_{fix}, v_{fix}, x_{fix} = \left\{ q_{fov}^{i,j}, k_{fov}^{i,j}, v_{fov}^{i,j}, x^{i,j} \mid g(i,j) = 1 \right\}. \tag{8}$$

In this architecture, the MHSA module forms a global relationship among fixated tokens and produces an attention matrix for each input token, focusing on specific fixated values. This module adapts to varying input sizes by processing solely the fixated tokens (where $d_h$ denotes the number of heads and $\sigma$ represents the softmax function):

$$\text{MHSA}(q_{fix}, k_{fix}, v_{fix}) = \sigma(q_{fix} k_{fix}^\top / \sqrt{d_h}) \cdot v_{fix} \tag{9}$$

FDT takes a full-size token map and produces foveated tokens, but since only fixated tokens are processed, their output shape ($x_{fix}$) does not align with the expected input size for subsequent blocks. To resolve this, we blend processed and unprocessed tokens, ensuring proper input for the next stages.

$$out^{i,j} = \begin{cases} x_{fix}^{i,j}, & \text{if} \quad g(i,j) = 1, \\ x^{i,j}, & otherwise. \end{cases} \tag{10}$$

### 2.4 TRAINING DYNAMIC NETWORK

In the FDT architecture, the fixation module functions as a gating network, selectively applying a selection operation to foveated tokens from the input image. Due to the diverse characteristics of each input, the number of selected tokens varies, complicating training in mini-batches. To facilitate mini-batch training, all foveated tokens are fed into the MHSA module without applying fixation sampling. For a differentiable fixation map enabling end-to-end training, we use Gumbel-Softmax with hard labeling on the output of the foveation module during training, instead of the logit comparison outlined in Equation 6.

$$x_{masked} = \text{MHSA}_{masked}(q_{fov}, k_{fov}, v_{fov}, g) + x,$$
$$x_{masked} = \text{MLP}(\text{LN}(x_{masked})) + x_{masked}. \tag{11}$$

The fixation map is then utilized for the calculation of masked attention (where $\mathcal{C}$ is a large constant):

$$\text{MHSA}_{masked}(q_{fov}, k_{fov}, v_{fov}, g) = \sigma(q_{fov} k_{fov}^{\top}/\sqrt{d_h} + \mathbb{M}) \cdot v_{fov},$$
$$\mathbb{M} = -\mathcal{C} \times (1 - flatten(g) \otimes flatten(g)). \tag{12}$$

**Budget Constraint.** In order to achieve high performance, the fixation module tends to assign all tokens as fixation points when there are no budget constraints. However, the ideal behavior of the fixation module should be to focus on the minimum number of tokens necessary for accurate prediction. To achieve this, we introduce a fixation budget constraint that forces the network to allocate a certain percentage of all tokens. The fixation budget loss is defined as the blockwise $\ell_2$ norm between the desired percentile and the mean of the fixation map to measure the deviation from the desired budget ($g$ is the fixation map, $\beta$ is the desired budget, and $L$ is the number of blocks):

$$\mathcal{L}_{Budget} = \frac{1}{L} \sum_{\ell=0}^{L-1} \|\mathbb{E}(g) - \beta\|^2, \tag{13}$$

Therefore, the total loss for training FDT on a classification task is the sum of cross-entropy loss as the task loss and the fixation budget loss ($\lambda$ is the balancing factor):

$$\mathcal{L} = \mathcal{L}_{CE} + \lambda \cdot \mathcal{L}_{Budget}. \tag{14}$$

## 3 EMPIRICAL ANALYSES

We demonstrate that FDT outperforms DeiT in robustness against adversarial attacks, natural corruption, and shortcut learning. We investigate the impact of budget size on both robustness and computational efficiency and provide attention and gating visualizations to support our findings. For experimental details, see Section A.2 in Appendix.

### 3.1 ADVERSARIAL ROBUSTNESS

When viewing an image perturbed by an adversarial attack, the brain may use fixation and foveation to focus on relevant features and ignore perturbations, potentially increasing robustness. To assess the robustness of FDT, we subjected it to 13 different adversarial attack methods. The results, shown in Figure 5, indicate that FDT outperformed the DeiT method in all types of attacks (for numerical results, see Table 5).

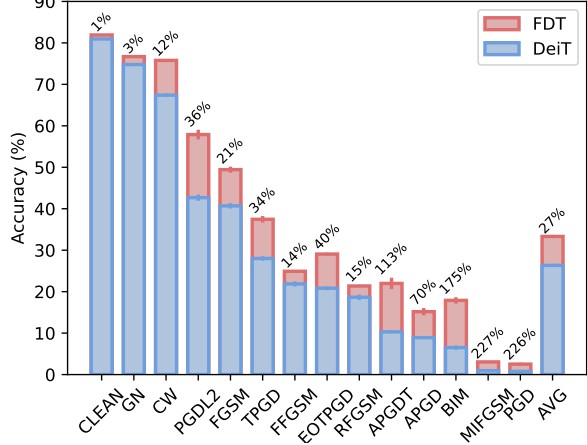

Figure 5: Adversarial robustness. FDT achieves a significant performance improvement, with an average increase of 27% compared to DeiT across all attack types (excluding clean data accuracy).

Table 1: Robustness to natural corruptions. Models evaluated across five severity levels using the ImageNet100-O dataset.

| | | DeiT | | | | | FDT | | | | |
|---|---|---|---|---|---|---|---|---|---|---|---|
| Corruption ↓ / Severity Level → | | 1 | 2 | 3 | 4 | 5 | 1 | 2 | 3 | 4 | 5 |
| Blur | Defocus Blur | $63.5_{\pm1.04}$ | $56.7_{\pm0.59}$ | $44.0_{\pm0.03}$ | $34.0_{\pm0.43}$ | $25.9_{\pm0.39}$ | $64.9_{\pm0.22}$ | $57.1_{\pm0.82}$ | $41.8_{\pm0.50}$ | $29.9_{\pm0.88}$ | $20.9_{\pm0.65}$ |
| | Glass Blur | $68.8_{\pm0.61}$ | $60.7_{\pm0.38}$ | $46.0_{\pm0.21}$ | $39.5_{\pm0.37}$ | $32.2_{\pm0.76}$ | $70.6_{\pm0.30}$ | $63.2_{\pm0.36}$ | $48.0_{\pm0.33}$ | $41.0_{\pm0.24}$ | $32.4_{\pm0.74}$ |
| | Motion Blur | $70.3_{\pm0.48}$ | $62.9_{\pm0.44}$ | $53.2_{\pm0.71}$ | $43.2_{\pm0.50}$ | $37.3_{\pm0.38}$ | $72.0_{\pm0.22}$ | $64.2_{\pm0.27}$ | $53.7_{\pm0.34}$ | $43.2_{\pm0.92}$ | $36.3_{\pm0.87}$ |
| | Zoom Blur | $63.0_{\pm0.58}$ | $57.4_{\pm1.12}$ | $53.7_{\pm0.95}$ | $49.7_{\pm0.60}$ | $45.6_{\pm0.76}$ | $62.6_{\pm0.75}$ | $56.0_{\pm0.54}$ | $51.8_{\pm0.44}$ | $47.6_{\pm0.27}$ | $42.9_{\pm0.64}$ |
| Digital | Contrast | $72.3_{\pm0.13}$ | $69.0_{\pm0.19}$ | $62.7_{\pm0.53}$ | $44.4_{\pm1.55}$ | $19.5_{\pm1.61}$ | $73.3_{\pm0.19}$ | $68.9_{\pm0.52}$ | $61.5_{\pm1.16}$ | $37.0_{\pm1.44}$ | $14.6_{\pm0.31}$ |
| | Elastic Transform | $75.2_{\pm0.44}$ | $67.2_{\pm0.21}$ | $74.7_{\pm0.53}$ | $72.1_{\pm0.48}$ | $62.5_{\pm0.36}$ | $76.9_{\pm0.30}$ | $67.0_{\pm0.18}$ | $76.2_{\pm0.12}$ | $74.0_{\pm0.39}$ | $64.7_{\pm0.93}$ |
| | JPEG Compression | $67.7_{\pm0.36}$ | $63.9_{\pm0.54}$ | $60.4_{\pm0.29}$ | $51.1_{\pm0.32}$ | $39.1_{\pm0.58}$ | $71.7_{\pm0.42}$ | $68.8_{\pm0.10}$ | $66.3_{\pm0.40}$ | $58.3_{\pm0.35}$ | $47.1_{\pm0.44}$ |
| | Pixelate | $77.3_{\pm0.33}$ | $76.3_{\pm0.32}$ | $71.5_{\pm0.71}$ | $59.9_{\pm1.42}$ | $48.6_{\pm3.35}$ | $78.8_{\pm0.38}$ | $78.1_{\pm0.13}$ | $73.7_{\pm0.17}$ | $63.8_{\pm0.35}$ | $52.3_{\pm0.62}$ |
| Noise | Gaussian Noise | $74.3_{\pm0.18}$ | $69.1_{\pm0.05}$ | $57.2_{\pm0.52}$ | $38.3_{\pm0.59}$ | $16.1_{\pm0.41}$ | $76.1_{\pm0.38}$ | $70.9_{\pm0.23}$ | $60.3_{\pm0.67}$ | $42.0_{\pm1.95}$ | $18.2_{\pm2.09}$ |
| | Impulse Noise | $71.8_{\pm0.05}$ | $63.8_{\pm0.26}$ | $55.8_{\pm0.59}$ | $35.0_{\pm0.63}$ | $15.8_{\pm0.36}$ | $73.9_{\pm0.31}$ | $66.9_{\pm0.59}$ | $59.2_{\pm1.40}$ | $39.0_{\pm1.93}$ | $18.0_{\pm2.06}$ |
| | Shot Noise | $74.2_{\pm0.22}$ | $67.6_{\pm0.32}$ | $55.4_{\pm0.54}$ | $32.7_{\pm0.60}$ | $18.4_{\pm0.75}$ | $76.1_{\pm0.21}$ | $69.7_{\pm0.36}$ | $58.8_{\pm0.91}$ | $36.4_{\pm1.92}$ | $21.1_{\pm2.00}$ |
| Weather | Brightness | $78.8_{\pm0.09}$ | $77.2_{\pm0.14}$ | $75.2_{\pm0.30}$ | $71.3_{\pm0.18}$ | $65.5_{\pm0.21}$ | $80.1_{\pm0.32}$ | $78.7_{\pm0.07}$ | $77.3_{\pm0.38}$ | $74.1_{\pm0.43}$ | $69.3_{\pm0.11}$ |
| | Fog | $68.0_{\pm0.19}$ | $62.4_{\pm0.44}$ | $53.5_{\pm0.35}$ | $49.4_{\pm1.15}$ | $38.3_{\pm1.34}$ | $70.2_{\pm0.21}$ | $64.8_{\pm0.48}$ | $56.0_{\pm0.45}$ | $52.4_{\pm0.24}$ | $43.1_{\pm0.48}$ |
| | Frost | $73.2_{\pm0.36}$ | $66.5_{\pm0.37}$ | $59.8_{\pm0.84}$ | $59.2_{\pm0.85}$ | $54.1_{\pm0.60}$ | $75.8_{\pm0.19}$ | $69.3_{\pm0.28}$ | $63.2_{\pm0.52}$ | $62.2_{\pm0.56}$ | $57.0_{\pm0.78}$ |
| | Snow | $66.9_{\pm0.56}$ | $52.0_{\pm0.28}$ | $53.7_{\pm0.42}$ | $44.7_{\pm0.22}$ | $41.9_{\pm0.53}$ | $69.0_{\pm0.24}$ | $55.4_{\pm0.46}$ | $57.9_{\pm0.81}$ | $48.8_{\pm0.68}$ | $46.6_{\pm0.12}$ |
| **Average** | | $71.0_{\pm0.20}$ | $64.8_{\pm0.11}$ | $58.5_{\pm0.19}$ | $48.3_{\pm0.21}$ | $37.4_{\pm0.43}$ | $72.8_{\pm0.10}$ | $66.6_{\pm0.03}$ | $60.4_{\pm0.18}$ | $50.0_{\pm0.38}$ | $39.0_{\pm0.43}$ |

## 3.2 NATURAL CORRUPTION ROBUSTNESS

We evaluated models using the ImageNet100-O dataset, which includes common corruptions. This subset of ImageNet-O contains the common classes of ImageNet100. Table 1 shows that FDT consistently outperforms DeiT across all severity levels, particularly in weather-related corruptions. This robustness makes FDT highly suitable for safety-critical applications where dependable visual perception is crucial, akin to the HVS.

## 3.3 ROBUSTNESS AGAINST SHORTCUT LEARNING

Shortcut learning occurs when ANNs form decision rules that excel on specific datasets by exploiting spurious correlations or statistical irregularities instead of learning the underlying task. These strategies, though effective on familiar data, do not generalize well across different data distributions (Geirhos et al., 2020). To assess models, we trained both DeiT and the FDT under varying budget constraints using the Tinted-ImageNet100 dataset, where each sample is modified with a class-specific tint, following the approach in the Tinted-STL10 dataset. Performance was evaluated using the standard ImageNet100 validation set. Figure 6 shows that FDT is more robust to shortcut learning than DeiT, especially under larger budgets, demonstrating its capability for robust generalization in diverse and challenging environments.

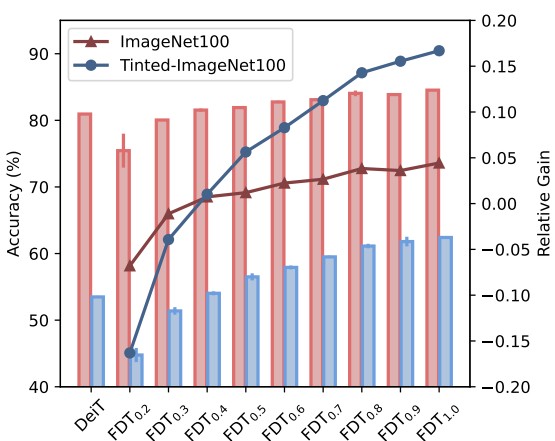

Figure 6: Robustness to shortcut learning. Actual performance of each model displayed as bars, with the relative gains of FDT over DeiT depicted as a line graph.

## 3.4 EFFECT OF BUDGET

Our model incorporates the MHSA module, which exhibits quadratic growth in computational complexity as the number of tokens increases. The fixation module within each FDT block dynamically selects a subset of tokens for processing in the MHSA based on the input image's complexity, significantly reducing the overall computational cost.

Table 2: Effect of gating budgets on computational efficiency and model performance. We report average accuracies for clean (Acc.), adversarially attacked (Adv.), and naturally corrupted (Corr.) samples, alongside clean accuracy for models trained on tinted samples (Shct.) as a robustness measure against shortcut learning. 'Eff. Fix.' denotes the effective fixation ratio on the validation dataset, with subscripts indicating FDT's fixation budget. FDT demonstrates high performance and computational efficiency, capitalizing on the minimal overhead of its foveation and fixation modules.

| | Accuracy | | | | | | Relative Gain | |
| | Acc. | Adv. | Corr. | Shct. | Eff. Fix. | GMAC | Acc. | GMAC |
|---|---|---|---|---|---|---|---|---|
| DeiT | $80.9_{\pm0.11}$ | $26.3_{\pm0.43}$ | $56.0_{\pm0.18}$ | $53.5_{\pm0.16}$ | $1.00_{\pm0.000}$ | $4.60_{\pm0.00}$ | 0 | 0 |
| $FDT_{0.2}$ | $75.4_{\pm2.55}$ | $\mathbf{45.9}_{\pm0.40}$ | $48.2_{\pm2.40}$ | $44.8_{\pm1.07}$ | $0.29_{\pm0.002}$ | $2.09_{\pm0.01}$ | +9.38 | −54.57 |
| $FDT_{0.3}$ | $80.0_{\pm0.06}$ | $40.1_{\pm1.71}$ | $54.5_{\pm0.28}$ | $51.4_{\pm0.56}$ | $0.37_{\pm0.004}$ | $2.36_{\pm0.01}$ | +11.19 | −48.70 |
| $FDT_{0.4}$ | $81.5_{\pm0.23}$ | $36.8_{\pm3.42}$ | $56.6_{\pm0.26}$ | $54.0_{\pm0.29}$ | $0.45_{\pm0.007}$ | $2.63_{\pm0.03}$ | +10.67 | −42.83 |
| $FDT_{0.5}$ | $81.9_{\pm0.05}$ | $33.3_{\pm0.45}$ | $57.8_{\pm0.17}$ | $56.5_{\pm0.53}$ | $0.56_{\pm0.006}$ | $3.01_{\pm0.02}$ | +9.17 | −34.57 |
| $FDT_{0.6}$ | $82.8_{\pm0.19}$ | $32.8_{\pm0.52}$ | $59.2_{\pm0.38}$ | $57.9_{\pm0.29}$ | $0.68_{\pm0.003}$ | $3.47_{\pm0.01}$ | +10.25 | −24.57 |
| $FDT_{0.7}$ | $83.1_{\pm0.17}$ | $31.8_{\pm0.23}$ | $59.9_{\pm0.30}$ | $59.5_{\pm0.19}$ | $0.81_{\pm0.004}$ | $3.93_{\pm0.01}$ | +10.45 | −14.57 |
| $FDT_{0.8}$ | $84.0_{\pm0.42}$ | $31.5_{\pm0.09}$ | $60.9_{\pm0.40}$ | $61.1_{\pm0.34}$ | $0.93_{\pm0.001}$ | $4.36_{\pm0.01}$ | +11.64 | −5.22 |
| $FDT_{0.9}$ | $83.9_{\pm0.18}$ | $32.4_{\pm0.08}$ | $61.8_{\pm0.40}$ | $61.8_{\pm0.72}$ | $0.99_{\pm0.001}$ | $4.59_{\pm0.00}$ | +13.19 | −0.22 |
| $FDT_{1.0}$ | $\mathbf{84.5}_{\pm0.07}$ | $32.5_{\pm0.68}$ | $\mathbf{62.9}_{\pm0.17}$ | $\mathbf{62.4}_{\pm0.18}$ | $1.00_{\pm0.000}$ | $4.63_{\pm0.00}$ | +14.25 | +0.65 |

To evaluate our model's efficiency, we quantified the Multiply-Accumulate operations (MACs) required for inference for both DeiT and FDT under various gating budgets. We calculated FDT's computational complexity by averaging the computations needed per sample in the validation set, reflecting the dynamic nature of our approach. MACs are normalized to DeiT to highlight computational efficiency gains. Table 2 shows that FDT requires fewer expected MACs for inference than DeiT. At a 50% budget, FDT uses an average of 3.01 GMACs, achieving a 34.57% reduction compared to DeiT. Even at a full budget, where all tokens are utilized, the increase in complexity is only 0.7%, demonstrating the minimal overhead of the foveation and fixation modules. These findings underscore FDT's ability to maintain high performance and computational efficiency, enhancing its robustness and making it suitable for a wide range of applications.

## 3.5 FEATURE INVERSION

We employ "feature inversion" to enhance our understanding and visualization of transformer-based representations. This technique reconstructs an input image from specific model features or activations, offering insights into how the model processes inputs and makes predictions. While widely used with CNNs (Simonyan et al., 2013; Selvaraju et al., 2017), its application to transformers remains less explored. Using the 'Deep Image Prior' method (Engstrom et al., 2019), we optimize a CNN-based network $F_\theta(\cdot)$ to transform random noise input $z$ into an image. The goal is to match the features of the output image, particularly the CLS features, with those of a target image $I$, defined by:

$$\arg\min_\theta \|\phi\left(F_\theta(z)\right) - \phi(I)\|_F \tag{15}$$

where $\phi(I)$ denotes the target features and $|.|_F$ represents the Frobenius norm, focusing on the classification token (CLS); thus, $\phi(I) = CLS(I)$.

We employ the same network architecture and parameters as Engstrom et al. (2019) for the generative network $F_\theta(z)$. The results in Figure 7 reveal clear distinctions between the reconstruction images of DeiT and FDT. Notably, FDT's reconstructions are more realistic and semantically richer than those of DeiT, supporting the hypothesis (Engstrom et al., 2019) that networks designed for robustness produce more semantically meaningful representations.

## 3.6 EFFECT OF MODEL SIZE

The relationship between model size and performance is pivotal in neural network design, balancing capacity with computational efficiency. Generally, larger models can achieve higher accuracy but may overfit and require more resources. Smaller models might better generalize and suit practical needs but have limited learning capacity for certain functions. Understanding this trade-off is crucial for choosing the appropriate model size for specific tasks.

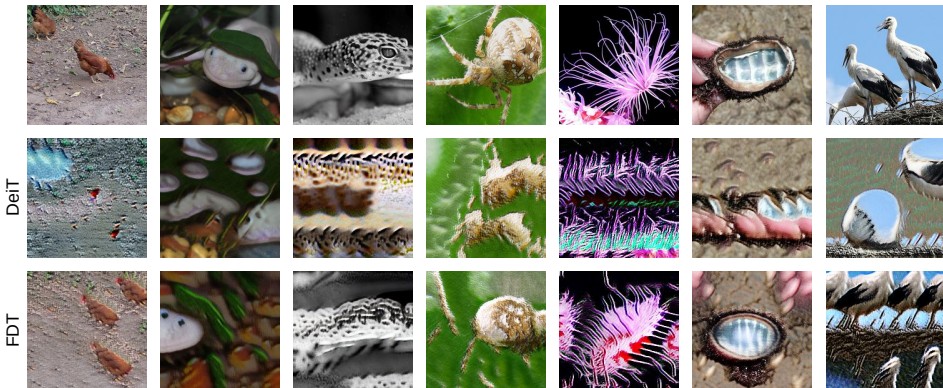

Figure 7: Feature inversion using 'Deep Image Prior' method for DeiT-S and FDT-S trained at a 50% budget. This comparison demonstrates that FDT learns more meaningful representations than DeiT.

Table 3: Comparison of FDT across different model sizes (tiny, small, and base) against the DeiT model in terms of accuracy on clean, adversarially attacked, and naturally corrupted images, as well as the relative accuracy gain over DeiT for all model sizes.

| Size | Model | GMAC | Acc. | Adv. Acc. | Corr. Acc. | Rel. Gain |
|------|-------|------|------|-----------|------------|-----------|
| Tiny | DeiT | $1.26_{\pm0.00}$ | $64.4_{\pm0.48}$ | $17.6_{\pm0.32}$ | $40.8_{\pm0.24}$ | 0 |
| | FDT | $0.84_{\pm0.00}$ | $67.1_{\pm0.08}$ | $24.7_{\pm1.67}$ | $43.5_{\pm0.15}$ | +17.05 |
| Small | DeiT | $4.60_{\pm0.00}$ | $80.9_{\pm0.11}$ | $26.3_{\pm0.43}$ | $56.0_{\pm0.18}$ | 0 |
| | FDT | $3.01_{\pm0.02}$ | $81.9_{\pm0.05}$ | $33.3_{\pm0.45}$ | $57.2_{\pm0.31}$ | +10.00 |
| Base | DeiT | $17.57_{\pm0.00}$ | $81.5_{\pm0.25}$ | $30.0_{\pm0.76}$ | $56.9_{\pm0.47}$ | 0 |
| | FDT | $11.56_{\pm0.07}$ | $82.2_{\pm0.28}$ | $40.8_{\pm1.80}$ | $57.8_{\pm0.17}$ | +12.81 |

We explore the impact of model size on our FDT by evaluating three sizes: tiny, small, and base, following DeiT conventions. We compared FDT's performance against DeiT across the same evaluation metrics, except for the learning rate of the base model set to 2e-4. Our results, presented in Table 3, cover accuracy on clean, adversarially attacked, and naturally corrupted images, and the relative accuracy gains over DeiT. Our findings indicate that FDT performs well across sizes, showing high accuracy on clean images and robustness to adversarial and natural challenges. FDT's adaptability to various computational demands, coupled with its efficiency and performance, makes it well-suited for diverse applications.

### 3.7 REACTION TIME ANALYSIS

RT in HVS measures the duration to respond to a visual stimulus, reflecting neural and cognitive processes in perception, attention, and decision-making, and varies with task complexity, object number and similarity, and uncertainties like occlusions or viewpoint changes, generally increasing with higher difficulty. Our study extends RT analysis to a neural network, comparing its performance to the HVS by measuring response times to visual stimuli and accuracy for samples classified as easy, medium, and hard (see Table 4), revealing an inverse relationship between sample complexity and accuracy. The model's RTs closely match those of the HVS, indicating it effectively mimics human foveation and fixation, suggesting its potential as a tool for studying human cognition by replicating intricate behaviors in visual perception and cognitive functions (see Figure 8).

### 3.8 EVALUATING MODEL DECISIONS: VISUALIZATION TECHNIQUES AND INSIGHTS

We employ visualizations of eye movements, fixation maps, and attention maps to mirror the human visual system's saccadic processing. The FDT uses a fixation module to generate fixation maps that simulate high-resolution visual acquisition through these movements, highlighting fixation probabilities within scenes. To demonstrate the FDT's simulated "eye movements", we identify the most likely token positions as fixation points. Figure 9 illustrates these points and their sequence

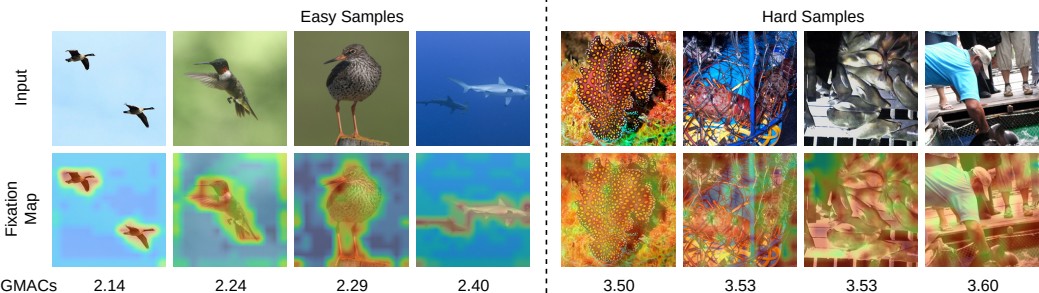

Figure 8: Visual comparison of easy and hard samples with their computational requirements for inference. The first row shows the original images, the second row shows heatmaps of fixation points, and the final row lists the GMACs required for each image. This highlights how computational load varies with the number and distribution of fixation points, reflecting inference efficiency and focus

|  |  |  | Easy | Medium | Hard |
|---|---|---|---|---|---|
| Clean |  |  | $84.9_{\pm0.35}$ | $80.5_{\pm0.25}$ | $80.3_{\pm0.31}$ |
| Adversarial Att. | PGD $\ell_2$ |  | $60.6_{\pm0.35}$ | $56.4_{\pm2.12}$ | $56.6_{\pm1.32}$ |
| Natural Corruption | Blur | Defocus B. | $45.9_{\pm0.81}$ | $43.8_{\pm0.72}$ | $39.1_{\pm0.94}$ |
| | | Glass B. | $55.7_{\pm0.71}$ | $52.0_{\pm0.06}$ | $45.5_{\pm0.07}$ |
| | | Motion B. | $60.0_{\pm0.24}$ | $54.4_{\pm0.36}$ | $47.2_{\pm1.33}$ |
| | | Zoom B. | $63.0_{\pm1.33}$ | $52.1_{\pm0.24}$ | $41.4_{\pm0.64}$ |
| | Digital | Contrast | $45.0_{\pm1.26}$ | $51.1_{\pm1.20}$ | $57.1_{\pm0.53}$ |
| | | Elastic Tran. | $75.4_{\pm0.31}$ | $71.4_{\pm0.37}$ | $68.5_{\pm0.44}$ |
| | | JPEG Comp. | $68.7_{\pm1.31}$ | $62.2_{\pm0.77}$ | $56.4_{\pm0.65}$ |
| | | Pixelate | $74.8_{\pm0.28}$ | $68.4_{\pm0.51}$ | $64.8_{\pm0.72}$ |
| | Noise | Gaussian N. | $54.4_{\pm1.20}$ | $52.2_{\pm1.07}$ | $54.0_{\pm1.59}$ |
| | | Impulse N. | $53.0_{\pm1.65}$ | $50.1_{\pm1.40}$ | $51.2_{\pm1.37}$ |
| | | Shot N. | $53.8_{\pm1.44}$ | $50.9_{\pm1.37}$ | $52.5_{\pm1.69}$ |
| | Weather | Brightness | $79.3_{\pm0.77}$ | $75.0_{\pm0.10}$ | $73.4_{\pm0.38}$ |
| | | Fog | $61.1_{\pm0.85}$ | $57.4_{\pm1.00}$ | $53.5_{\pm0.35}$ |
| | | Frost | $70.6_{\pm0.54}$ | $64.3_{\pm0.95}$ | $61.7_{\pm0.43}$ |
| | | Snow | $63.0_{\pm0.46}$ | $54.6_{\pm0.40}$ | $49.1_{\pm1.20}$ |

Table 4: Accuracy metrics across samples of varying difficulty levels, subjected to different noise types. The validation dataset was divided into three balanced subsets based on average fixations per sample: easy' for the fewest, medium' for an intermediate number, and 'hard' for the most fixations. For analyses of natural corruption, samples from all severity levels were included.

(using arrows and color transitions from green to yellow to indicate the sequence of fixations; and marking fixation points with circles and blurring regions outside of fixation for clarity). The system's focus intensity varies based on the visual input, adapting to scenes with multiple objects through diverse sampling. Using the Gumbel-Softmax and hard label techniques, we determine tokens as fixation points based on their likelihood values, producing binary maps that indicate these points. Averaging these maps across all blocks, we create overall fixation maps shown in Figure 9, which depict the model's strategy to maximize informative content while minimizing irrelevant background areas. Additionally, we visualize attention towards the classification token to understand decision processes within vision transformers, using the attention rollout method (Abnar & Zuidema, 2020). Figure 9 demonstrates that while the fixation module covers broad informative areas, the attention maps concentrate on the most discriminative regions for class identification. This confirms our model's effectiveness in mimicking human visual attention mechanisms.

## 4 CONCLUSION

We introduced the *Foveated Dynamic Vision Transformer (FDT)*, a novel architecture inspired by human visual system mechanisms, enhancing computational efficiency and robustness against adversarial attacks, natural corruption, and shortcut learning. Our results on the ImageNet100 dataset demonstrate FDT's superior accuracy, efficiency, and robustness compared to the baseline DeiT-Small architecture. This research contributes to biologically inspired computational models, integrating human visual principles into deep learning architectures. The FDT balances high performance with

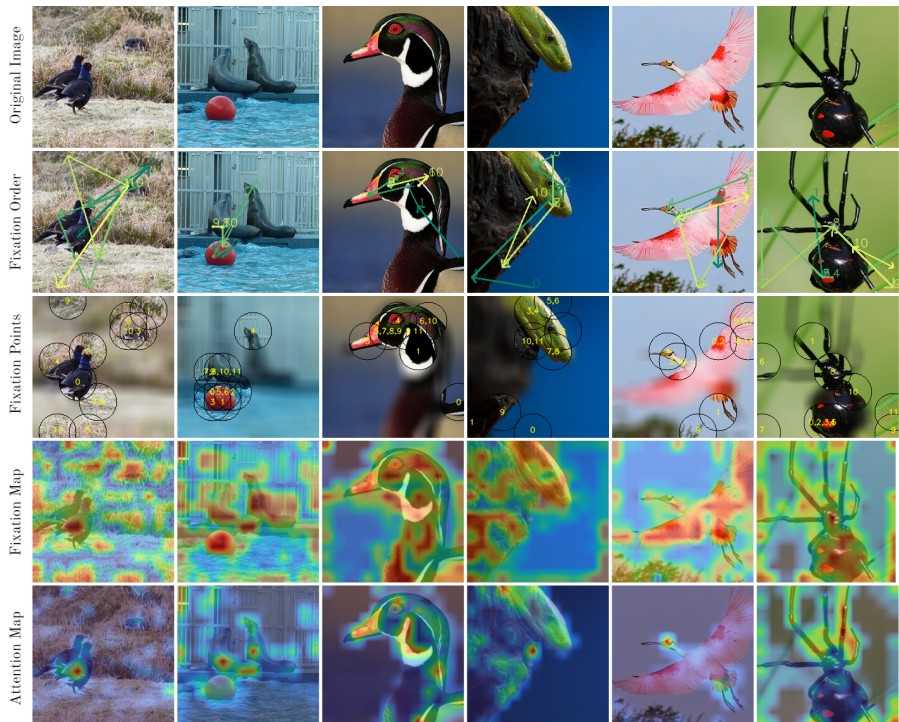

Figure 9: Visualization of fixation points and their orders, fixation maps, and attention maps. Each column displays diverse samples from the ImageNet100 validation set.

computational efficiency, making it suitable for resource-constrained environments and highlighting its capability to focus on informative image regions, akin to human saccadic movements. Future work could extend FDT to domains like video processing and augmented reality, where dynamic foveation can reduce computational demands while maintaining performance, and explore its deployment in real-world, resource-limited scenarios, underscoring the value of bioinspired approaches in developing efficient, robust AI models.

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

# A APPENDIX

## A.1 RELATED WORK

Several approaches inspired by the functioning of the human visual system have been devised to replicate foveation and eye movement within the domain of computer vision. Mnih et al. (2014) introduced a method that employs a sequence of image movements to aggregate information before classification. This method utilizes a hard-attention mechanism implemented through reinforcement learning to predict fixation points. In a similar vein, Akbas & Eckstein (2017) developed a foveated object detection system that harnesses varying resolutions to comprehensively analyze the entire image, aligning its fovea with regions of interest in the input data. This approach amalgamates data from multiple fixations and leverages peripheral information, similar to the way the human visual system employs contextual cues to guide gaze.

Furthermore, Deza & Konkle (2020) explored the impact of foveation on machine vision employing two-stage models, encompassing an image transformation stage and a deep convolutional neural network. Their investigation primarily focused on evaluating the effects of texture-based coding in the visual periphery on subsequent visual representations. Thavamani et al. (2021) proposed a foveated object detector that utilizes an image-magnification approach, preserving high resolution for points of interest while maintaining a compact canvas size.

Lukanov et al. (2021) developed an efficient model that incorporates space-variant sampling, mimicking the human retina, and the mechanisms to generate sequences of fixations. They proposed a CNN-based method that uses Foveal Cartesian Geometry (FCG) sampling, as outlined by Martínez & Robles (2006), to compress visual signals. An attention mechanism is employed for "eye movements" to progressively gather detailed information from a scene. Activation within the feature maps of the final convolutional layer is harnessed to guide the generation of fixation sequences. Additionally, Wang et al. (2021) investigate the role of foveation and saccadic eye movements as biologically inspired proxies for data augmentation in the context of self-supervised learning (SSL). They suggest that foveation through cortical magnification and saccade-like sampling of images can replace conventional SSL augmentations, offering insights into potential biological implementations of self-supervision and challenging spatially uniform processing assumptions in both human and machine vision.

These approaches have demonstrated promising results in emulating the foveation and eye movement processes of the human visual system, with potential applications across a variety of computer vision tasks. However, it is imperative to emphasize the critical necessity for further research, not only aimed at optimizing and refining the functionality of the human visual system, particularly in the context of vision transformers, but also unlocking its transformative potential for artificial intelligence, ultimately advancing the frontiers of machine perception and understanding. To this end, we aim to design a system that effectively mimics the human visual system's process of selectively focusing on certain regions of an image and using surrounding context to guide the gaze.

## A.2 EXPERIMENTAL SETTINGS

To evaluate FDT, we conduct image classification experiments on the ImageNet-100 dataset, a subset of the ImageNet-1k dataset containing 100 randomly chosen classes with 1300 training images and 50 validation images per class. We report top-1 accuracy using a single 224x224 crop and compare FDT to DeiT (Touvron et al., 2021) using the same training settings, including AdamW optimization for 300 epochs with cosine learning rate decay and 20 epochs of linear warm-up, batch size of 512, and V100 GPUs for training and evaluation. The initial learning rate, weight decay, and momentum are set to 0.001, 0.05, and 0.9, respectively. Unless otherwise stated, all experiments use a budget of $\beta = 0.5$. All results are reported as the mean and one standard deviation of three differently initialized runs. For training Tiny, Small, and Base models, one, two, and four V100 GPUs with 32GB of memory are used, respectively. Evaluations of the models are done on a single V100 GPU.

**Adversarial Robustness.** The adversarial attack methods used to assess the robustness of FDT include gaussian noise (GN), CW (Carlini & Wagner, 2017), PGD(L2) (Madry et al., 2017), FGSM (Goodfellow et al., 2014), TPGD (Zhang et al., 2019), FFGSM (Wong et al., 2020), EOTPGD (Liu et al., 2018), RFGSM (Tramèr et al., 2017), APGD(T) (Croce & Hein, 2020), BIM (Kurakin

et al., 2018), and MIFGSM (Dong et al., 2018). We used the TorchAttacks library Kim (2020) for implementing the attacks. All methods were run with default values, except for the Projected Gradient Descent (PGD) method, which had an epsilon value of 0.1. As demonstrated in Table 5 in the Appendix, although a lower budget results in some accuracy drops, it overall helps to produce a more robust model. These findings support that the FDT method effectively enhances robustness against adversarial attacks by focusing on relevant features and filtering out noise.

## A.3 ADVERSARIAL ATTACK RESULTS

Table 5: Comparison of robustness against 13 common adversarial attack methods for FDT and DeiT models trained on ImageNet-100 dataset. The models are labeled with their corresponding fixation budget hyperparameter (subscript) and model size (T, S, B for tiny, small, and base, respectively). The mean and one standard deviation of three runs with different initializations are reported.

| Attack Type | DeiT-T | DeiT-S | DeiT-B | FDT-T$_{0.5}$ | FDT-S$_{0.2}$ | FDT-S$_{0.3}$ | FDT-S$_{0.4}$ | FDT-S$_{0.5}$ | FDT-S$_{0.6}$ | FDT-S$_{0.7}$ | FDT-S$_{0.8}$ | FDT-S$_{0.9}$ | FDT-S$_{1.0}$ | FDT-B$_{0.5}$ |
|---|---|---|---|---|---|---|---|---|---|---|---|---|---|---|
| CLEAN | 64.4$_{\pm0.48}$ | 80.9$_{\pm0.11}$ | 81.5$_{\pm0.25}$ | 67.1$_{\pm0.08}$ | 75.4$_{\pm2.55}$ | 80.0$_{\pm0.06}$ | 81.5$_{\pm0.23}$ | 81.9$_{\pm0.05}$ | 82.8$_{\pm0.19}$ | 83.1$_{\pm0.17}$ | 84.0$_{\pm0.42}$ | 83.9$_{\pm0.18}$ | 84.5$_{\pm0.07}$ | 82.2$_{\pm0.28}$ |
| APGD | 16.8$_{\pm0.37}$ | 8.9$_{\pm0.11}$ | 8.8$_{\pm0.11}$ | 23.0$_{\pm2.50}$ | 46.7$_{\pm3.07}$ | 30.4$_{\pm3.78}$ | 22.0$_{\pm5.65}$ | 15.2$_{\pm0.88}$ | 13.3$_{\pm0.37}$ | 10.8$_{\pm1.38}$ | 8.9$_{\pm0.85}$ | 8.0$_{\pm0.19}$ | 7.1$_{\pm0.41}$ | 23.7$_{\pm3.03}$ |
| APGDT | 17.5$_{\pm0.25}$ | 10.3$_{\pm0.36}$ | 11.9$_{\pm0.28}$ | 27.1$_{\pm3.79}$ | 54.6$_{\pm1.00}$ | 40.0$_{\pm3.32}$ | 31.1$_{\pm6.44}$ | 22.0$_{\pm1.37}$ | 18.3$_{\pm1.12}$ | 14.4$_{\pm1.46}$ | 12.2$_{\pm0.82}$ | 10.6$_{\pm0.67}$ | 9.6$_{\pm0.46}$ | 34.7$_{\pm2.85}$ |
| BIM | 1.1$_{\pm0.12}$ | 6.5$_{\pm0.53}$ | 11.0$_{\pm0.43}$ | 8.3$_{\pm2.46}$ | 41.5$_{\pm0.83}$ | 30.8$_{\pm3.08}$ | 24.3$_{\pm5.84}$ | 17.9$_{\pm0.76}$ | 16.7$_{\pm0.47}$ | 12.2$_{\pm0.75}$ | 11.0$_{\pm0.72}$ | 12.2$_{\pm0.66}$ | 11.8$_{\pm1.38}$ | 32.2$_{\pm3.78}$ |
| CW | 55.1$_{\pm0.48}$ | 67.4$_{\pm0.42}$ | 71.4$_{\pm0.61}$ | 62.4$_{\pm1.08}$ | 74.2$_{\pm2.31}$ | 77.2$_{\pm0.62}$ | 77.1$_{\pm1.10}$ | 75.8$_{\pm0.31}$ | 75.5$_{\pm0.50}$ | 74.0$_{\pm0.46}$ | 73.2$_{\pm0.26}$ | 73.0$_{\pm0.26}$ | 72.2$_{\pm0.42}$ | 77.8$_{\pm0.38}$ |
| EOTPGD | 6.2$_{\pm0.13}$ | 20.8$_{\pm0.47}$ | 28.5$_{\pm1.40}$ | 14.6$_{\pm1.62}$ | 38.7$_{\pm0.60}$ | 34.2$_{\pm1.93}$ | 31.8$_{\pm2.91}$ | 29.1$_{\pm0.34}$ | 28.7$_{\pm0.92}$ | 28.5$_{\pm0.55}$ | 28.9$_{\pm1.08}$ | 30.4$_{\pm0.63}$ | 31.5$_{\pm1.21}$ | 35.3$_{\pm0.90}$ |
| FFGSM | 6.6$_{\pm0.39}$ | 21.9$_{\pm0.62}$ | 26.6$_{\pm1.84}$ | 11.4$_{\pm0.83}$ | 32.0$_{\pm0.75}$ | 27.6$_{\pm1.16}$ | 26.4$_{\pm1.88}$ | 24.9$_{\pm0.08}$ | 26.3$_{\pm1.20}$ | 28.1$_{\pm0.51}$ | 29.2$_{\pm0.75}$ | 32.8$_{\pm0.48}$ | 34.7$_{\pm1.24}$ | 32.6$_{\pm1.93}$ |
| FGSM | 18.8$_{\pm0.81}$ | 40.7$_{\pm0.65}$ | 45.4$_{\pm1.18}$ | 31.5$_{\pm2.78}$ | 59.2$_{\pm2.18}$ | 55.3$_{\pm1.69}$ | 52.6$_{\pm3.74}$ | 49.4$_{\pm0.73}$ | 48.8$_{\pm1.50}$ | 48.1$_{\pm0.59}$ | 48.2$_{\pm0.26}$ | 50.2$_{\pm0.46}$ | 50.9$_{\pm0.83}$ | 56.5$_{\pm1.71}$ |
| GN | 56.0$_{\pm0.42}$ | 74.8$_{\pm0.33}$ | 76.5$_{\pm0.40}$ | 60.1$_{\pm0.35}$ | 69.7$_{\pm2.34}$ | 74.4$_{\pm0.28}$ | 76.4$_{\pm0.34}$ | 76.7$_{\pm0.02}$ | 77.6$_{\pm0.39}$ | 78.6$_{\pm0.09}$ | 78.9$_{\pm0.43}$ | 79.4$_{\pm0.22}$ | 80.2$_{\pm0.22}$ | 77.7$_{\pm0.21}$ |
| MIFGSM | 0.1$_{\pm0.02}$ | 0.9$_{\pm0.26}$ | 1.7$_{\pm0.16}$ | 1.1$_{\pm0.36}$ | 16.6$_{\pm0.95}$ | 8.9$_{\pm2.21}$ | 5.7$_{\pm2.55}$ | 3.1$_{\pm0.07}$ | 3.0$_{\pm0.17}$ | 2.1$_{\pm0.16}$ | 1.9$_{\pm0.13}$ | 2.4$_{\pm0.06}$ | 2.5$_{\pm0.28}$ | 9.2$_{\pm2.15}$ |
| PGD | 0.1$_{\pm0.01}$ | 0.8$_{\pm0.32}$ | 1.6$_{\pm0.09}$ | 1.2$_{\pm0.38}$ | 18.4$_{\pm1.74}$ | 9.7$_{\pm2.29}$ | 6.1$_{\pm3.64}$ | 2.5$_{\pm0.35}$ | 2.3$_{\pm0.06}$ | 1.5$_{\pm0.05}$ | 1.4$_{\pm0.38}$ | 1.6$_{\pm0.16}$ | 1.3$_{\pm0.28}$ | 9.4$_{\pm2.83}$ |
| PGDL2 | 20.5$_{\pm1.19}$ | 42.7$_{\pm0.75}$ | 50.3$_{\pm0.86}$ | 40.3$_{\pm3.49}$ | 63.8$_{\pm2.37}$ | 62.6$_{\pm2.13}$ | 60.7$_{\pm4.03}$ | 57.9$_{\pm1.16}$ | 57.8$_{\pm1.34}$ | 55.5$_{\pm0.43}$ | 54.8$_{\pm0.26}$ | 55.2$_{\pm0.24}$ | 53.2$_{\pm1.29}$ | 64.8$_{\pm1.60}$ |
| RFGSM | 5.1$_{\pm0.18}$ | 18.7$_{\pm0.65}$ | 23.2$_{\pm2.16}$ | 9.0$_{\pm0.39}$ | 27.6$_{\pm0.75}$ | 24.2$_{\pm1.42}$ | 22.7$_{\pm1.94}$ | 21.4$_{\pm0.28}$ | 22.6$_{\pm1.04}$ | 24.3$_{\pm0.42}$ | 25.4$_{\pm0.90}$ | 28.7$_{\pm0.59}$ | 31.0$_{\pm1.40}$ | 28.3$_{\pm1.57}$ |
| TPGD | 25.1$_{\pm0.51}$ | 28.0$_{\pm0.53}$ | 32.6$_{\pm1.01}$ | 31.5$_{\pm1.90}$ | 54.3$_{\pm0.47}$ | 46.2$_{\pm2.03}$ | 41.3$_{\pm4.65}$ | 37.5$_{\pm0.79}$ | 35.9$_{\pm0.60}$ | 35.1$_{\pm0.58}$ | 35.0$_{\pm0.41}$ | 36.3$_{\pm0.47}$ | 36.6$_{\pm1.03}$ | 48.5$_{\pm2.04}$ |

Table 6: Comparison of robustness against 13 common adversarial attack methods for FDT and DeiT models trained on Tinted ImageNet-100 dataset. The models are labeled with their corresponding fixation budget hyperparameter (subscript) and model size (T, S, B for tiny, small, and base, respectively). The mean and one standard deviation of three runs with different initializations are reported.

| Attack Type | DeiT-S | FDT-S$_{0.2}$ | FDT-S$_{0.3}$ | FDT-S$_{0.4}$ | FDT-S$_{0.5}$ | FDT-S$_{0.6}$ | FDT-S$_{0.7}$ | FDT-S$_{0.8}$ | FDT-S$_{0.9}$ | FDT-S$_{1.0}$ |
|---|---|---|---|---|---|---|---|---|---|---|
| CLEAN | 53.5±0.16 | 44.8±1.07 | 51.4±0.56 | 54.0±0.29 | 56.5±0.53 | 57.9±0.29 | 59.5±0.19 | 61.1±0.34 | 61.8±0.72 | 62.4±0.18 |
| APGD | 24.9±1.27 | 33.0±1.43 | 28.5±1.45 | 26.1±1.74 | 25.1±1.10 | 23.0±0.77 | 22.0±1.15 | 22.0±1.09 | 21.5±0.54 | 22.5±1.39 |
| APGDT | 27.5±1.08 | 36.6±1.45 | 33.4±1.67 | 30.3±2.11 | 28.7±1.38 | 26.1±0.93 | 24.5±1.57 | 24.4±0.73 | 24.3±0.72 | 25.2±1.33 |
| BIM | 6.7±0.31 | 21.6±2.78 | 16.8±1.05 | 12.5±1.80 | 9.8±0.79 | 7.1±0.45 | 5.8±0.65 | 6.5±0.27 | 6.1±0.11 | 7.4±0.29 |
| CW | 50.6±0.48 | 44.1±1.02 | 50.0±0.63 | 52.2±0.76 | 54.1±0.83 | 55.2±0.26 | 56.2±0.67 | 57.5±0.23 | 58.6±0.62 | 58.9±0.43 |
| EOTPGD | 12.9±0.69 | 23.1±1.82 | 19.1±1.12 | 18.8±0.47 | 17.2±0.77 | 16.3±0.35 | 16.3±0.11 | 17.1±0.23 | 17.7±0.54 | 18.5±0.38 |
| FFGSM | 13.1±0.32 | 18.3±2.07 | 16.4±0.70 | 16.0±0.68 | 15.6±0.23 | 16.0±0.48 | 16.7±0.51 | 18.0±0.86 | 18.0±0.64 | 19.4±0.48 |
| FGSM | 24.4±0.27 | 32.6±1.88 | 33.0±0.61 | 31.1±1.16 | 29.6±0.70 | 28.9±0.34 | 28.6±0.20 | 30.0±0.35 | 30.6±0.80 | 30.9±0.11 |
| GN | 52.2±0.08 | 46.2±1.10 | 51.4±0.51 | 53.9±0.22 | 55.6±0.60 | 57.8±0.70 | 58.8±0.27 | 60.5±0.43 | 61.2±0.34 | 61.9±0.42 |
| MIFGSM | 1.1±0.14 | 8.0±2.12 | 3.8±0.56 | 2.1±0.26 | 1.4±0.18 | 1.1±0.21 | 1.1±0.31 | 1.1±0.07 | 1.2±0.16 | 2.6±0.23 |
| PGD | 1.0±0.17 | 9.4±2.36 | 3.8±0.58 | 2.2±0.36 | 1.3±0.22 | 0.8±0.13 | 0.6±0.17 | 0.7±0.08 | 0.7±0.09 | 1.5±0.16 |
| PGDL2 | 29.0±0.46 | 34.7±1.58 | 36.5±0.75 | 35.9±0.74 | 34.9±0.67 | 33.9±0.79 | 33.1±0.81 | 34.5±0.70 | 34.9±0.38 | 34.9±0.64 |
| RFGSM | 11.3±0.17 | 16.1±2.09 | 14.1±0.69 | 13.8±0.71 | 13.4±0.40 | 14.1±0.48 | 14.2±0.50 | 15.7±0.85 | 15.9±0.96 | 17.0±0.62 |
| TPGD | 25.4±1.17 | 36.5±1.57 | 34.4±1.36 | 31.8±1.11 | 29.5±1.11 | 29.1±0.25 | 29.7±0.94 | 29.0±0.40 | 30.3±0.42 | 32.3±0.48 |

## A.4 NATURAL CORRUPTION RESULTS

Table 7: Comparison of robustness against natural corruptions at five different severities for FDT and DeiT models trained on ImageNet-100 dataset. Models are evaluated using the ImageNet100-O dataset. The models are labeled with their corresponding fixation budget hyperparameter (subscript) and model size (T, S, B for tiny, small, and base, respectively). The mean and one standard deviation of three runs with different initializations are reported.

| Corruption | | | S | DeiT-T | DeiT-S | DeiT-B | FDT-T$_{0.5}$ | FDT-S$_{0.2}$ | FDT-S$_{0.3}$ | FDT-S$_{0.4}$ | FDT-S$_{0.5}$ | FDT-S$_{0.6}$ | FDT-S$_{0.7}$ | FDT-S$_{0.8}$ | FDT-S$_{0.9}$ | FDT-S$_{1.0}$ | FDT-B$_{0.5}$ |
|---|---|---|---|---|---|---|---|---|---|---|---|---|---|---|---|---|---|
| – | | | 0 | 64.4±0.48 | 80.9±0.11 | 81.5±0.25 | 67.1±0.08 | 75.4±2.55 | 80.0±0.06 | 81.5±0.23 | 81.9±0.05 | 82.8±0.19 | 83.1±0.17 | 84.0±0.42 | 83.9±0.18 | 84.5±0.07 | 82.2±0.28 |
| | | Defocus Blur | 1 | 44.6±0.05 | 63.5±1.04 | 62.8±0.85 | 46.6±0.28 | 50.2±3.75 | 60.5±0.65 | 62.5±0.59 | 64.9±0.22 | 66.0±0.12 | 67.0±0.64 | 68.1±0.50 | 68.5±0.65 | 69.6±0.51 | 62.9±1.00 |
| | | | 2 | 38.1±0.40 | 56.7±0.59 | 56.4±1.03 | 38.8±0.48 | 39.1±3.27 | 51.8±1.04 | 54.3±0.31 | 57.1±0.82 | 58.8±0.38 | 60.1±0.65 | 61.7±0.97 | 62.6±1.17 | 63.5±0.58 | 55.3±0.79 |
| | | | 3 | 28.4±0.33 | 44.0±0.03 | 44.7±1.00 | 27.4±0.25 | 21.0±4.10 | 33.7±1.63 | 37.2±0.44 | 41.8±0.50 | 45.2±1.26 | 47.2±0.73 | 50.5±1.01 | 51.1±1.36 | 51.7±0.68 | 41.0±0.50 |
| | | | 4 | 21.6±0.49 | 34.0±0.43 | 34.5±1.11 | 20.0±0.27 | 12.2±3.51 | 20.9±1.44 | 24.3±0.78 | 29.9±0.88 | 33.4±0.92 | 36.5±1.07 | 39.4±0.42 | 40.4±1.20 | 40.2±1.14 | 28.9±0.64 |
| | | | 5 | 17.2±0.64 | 25.9±0.39 | 26.2±0.73 | 14.9±0.82 | 7.7±2.78 | 13.7±0.67 | 16.0±0.74 | 20.9±0.65 | 23.8±0.53 | 26.4±1.19 | 29.8±0.34 | 30.9±0.82 | 30.4±1.37 | 20.5±0.79 |
| | | | 1 | 51.9±0.15 | 68.8±0.61 | 68.4±0.38 | 55.0±0.04 | 60.4±4.85 | 68.3±0.14 | 70.0±0.43 | 70.6±0.30 | 71.8±0.21 | 72.0±0.52 | 72.4±0.05 | 72.7±0.44 | 73.3±0.44 | 69.2±0.86 |

Glass Blur

Blur

Table 7: Comparison of robustness against natural corruptions at five different severities for FDT and DeiT models trained on ImageNet-100 dataset. Models are evaluated using the ImageNet100-O dataset. The models are labeled with their corresponding fixation budget hyperparameter (subscript) and model size (T, S, B for tiny, small, and base, respectively). The mean and one standard deviation of three runs with different initializations are reported.

| Corruption | | S | DeiT-T | DeiT-S | DeiT-B | FDT-T$_{0.5}$ | FDT-S$_{0.2}$ | FDT-S$_{0.3}$ | FDT-S$_{0.4}$ | FDT-S$_{0.5}$ | FDT-S$_{0.6}$ | FDT-S$_{0.7}$ | FDT-S$_{0.8}$ | FDT-S$_{0.9}$ | FDT-S$_{1.0}$ | FDT-B$_{0.5}$ |
|---|---|---|---|---|---|---|---|---|---|---|---|---|---|---|---|---|
| | | 2 | 44.5±0.33 | 60.7±0.38 | 60.5±0.83 | 47.0±0.47 | 51.0±5.05 | 59.4±0.49 | 61.7±0.71 | 63.2±0.36 | 63.9±0.40 | 64.2±0.51 | 64.6±0.04 | 64.8±1.12 | 66.1±0.51 | 60.6±0.81 |
| | | 3 | 36.2±0.72 | 46.0±0.21 | 45.9±0.93 | 36.2±0.54 | 37.4±5.23 | 44.7±0.76 | 46.4±0.83 | 48.0±0.33 | 48.0±0.42 | 48.2±0.62 | 48.3±0.20 | 48.7±1.09 | 49.5±0.34 | 46.1±0.99 |
| | | 4 | 31.2±0.74 | 39.5±0.37 | 39.6±1.02 | 30.6±0.69 | 30.7±4.49 | 36.9±1.09 | 39.4±0.72 | 41.0±0.24 | 40.9±0.26 | 41.3±0.44 | 41.8±0.42 | 41.7±1.17 | 42.5±0.22 | 39.5±0.82 |
| | | 5 | 23.8±0.55 | 32.2±0.76 | 32.2±0.70 | 22.5±0.53 | 20.8±3.03 | 26.7±0.08 | 29.6±0.43 | 32.4±0.74 | 32.2±0.51 | 33.2±0.17 | 34.1±0.65 | 34.0±0.96 | 35.0±0.19 | 30.8±0.49 |
| | Motion Blur | 1 | 52.9±0.22 | 70.3±0.48 | 71.0±0.20 | 55.6±0.74 | 61.6±3.17 | 69.3±0.10 | 70.6±0.43 | 72.0±0.22 | 73.0±0.15 | 73.5±0.36 | 74.6±0.07 | 75.4±0.20 | 75.6±0.30 | 71.3±0.88 |
| | | 2 | 45.7±0.20 | 62.9±0.44 | 64.1±0.30 | 48.0±0.60 | 51.1±2.84 | 61.0±0.51 | 62.2±0.07 | 64.2±0.27 | 65.6±0.02 | 66.4±0.67 | 68.1±0.28 | 68.8±0.32 | 69.2±0.74 | 63.8±0.77 |
| | | 3 | 37.5±0.61 | 53.2±0.71 | 54.3±0.63 | 38.7±0.90 | 38.9±2.28 | 49.6±0.84 | 51.3±0.28 | 53.7±0.34 | 55.9±0.24 | 57.1±0.74 | 58.2±0.19 | 59.7±0.55 | 59.9±1.07 | 53.8±0.86 |
| | | 4 | 30.1±0.78 | 43.2±0.50 | 43.6±0.80 | 30.6±0.83 | 27.2±2.83 | 38.2±0.62 | 39.9±0.24 | 43.2±0.92 | 44.8±0.23 | 45.8±0.72 | 47.2±0.39 | 48.1±0.54 | 48.3±1.44 | 42.1±0.49 |
| | | 5 | 26.5±1.21 | 37.3±0.38 | 37.2±0.91 | 26.3±0.71 | 21.5±3.04 | 31.8±0.54 | 33.4±0.05 | 36.3±0.87 | 37.9±0.10 | 38.9±0.63 | 40.4±0.56 | 41.2±0.57 | 40.6±1.40 | 35.4±0.40 |
| | Zoom Blur | 1 | 48.4±0.07 | 63.0±0.58 | 63.1±0.77 | 49.8±0.19 | 53.7±3.05 | 61.0±0.66 | 61.9±0.52 | 62.6±0.75 | 63.3±0.35 | 64.0±0.18 | 64.1±0.53 | 64.2±0.27 | 65.0±0.52 | 61.4±0.52 |
| | | 2 | 43.3±0.21 | 57.4±1.12 | 57.4±0.60 | 44.1±0.52 | 46.5±2.50 | 54.0±0.67 | 54.9±0.38 | 56.0±0.54 | 56.9±0.15 | 57.7±0.36 | 57.5±0.54 | 57.8±0.26 | 58.3±0.37 | 55.2±0.94 |
| | | 3 | 40.1±0.21 | 53.7±0.95 | 53.5±1.02 | 40.5±1.14 | 41.0±2.86 | 49.3±0.46 | 50.9±0.10 | 51.8±0.44 | 53.0±0.17 | 53.6±0.27 | 53.8±0.34 | 54.2±0.28 | 54.4±0.70 | 51.6±0.95 |
| | | 4 | 37.1±0.54 | 49.7±0.60 | 49.5±0.73 | 37.2±0.79 | 36.4±2.92 | 45.4±0.36 | 46.4±0.47 | 47.6±0.27 | 48.5±0.48 | 49.1±0.17 | 49.1±0.53 | 49.8±0.39 | 49.9±0.76 | 47.4±0.65 |
| | | 5 | 34.1±0.80 | 45.6±0.76 | 45.5±0.63 | 33.7±0.84 | 31.7±3.54 | 41.3±0.34 | 41.6±0.40 | 42.9±0.64 | 44.1±0.54 | 44.7±0.29 | 44.8±0.45 | 45.0±0.22 | 44.9±0.85 | 43.4±0.79 |
| Digital | Contrast | 1 | 53.4±0.37 | 72.3±0.13 | 73.5±0.41 | 55.8±0.24 | 56.9±5.51 | 68.9±0.32 | 72.1±0.37 | 73.3±0.19 | 75.1±0.42 | 75.3±1.19 | 76.6±0.54 | 76.8±0.62 | 77.6±0.33 | 74.4±0.38 |
| | | 2 | 48.7±0.90 | 69.0±0.19 | 70.0±0.90 | 49.7±0.90 | 47.5±4.11 | 61.2±1.43 | 67.3±0.47 | 68.9±0.52 | 70.8±0.77 | 72.2±1.77 | 73.0±0.81 | 73.4±0.70 | 75.1±0.26 | 70.1±0.41 |
| | | 3 | 41.2±0.90 | 62.7±0.53 | 63.7±1.20 | 39.3±1.31 | 32.4±0.80 | 46.0±2.19 | 56.5±1.32 | 61.5±1.16 | 64.1±0.98 | 66.7±2.05 | 69.1±1.00 | 69.3±0.76 | 71.1±0.29 | 60.6±0.49 |
| | | 4 | 24.9±0.90 | 44.4±1.55 | 46.4±2.21 | 20.1±1.14 | 13.3±1.48 | 21.4±1.49 | 32.0±1.68 | 37.0±1.44 | 42.9±1.96 | 46.2±2.41 | 53.8±0.33 | 53.1±1.27 | 58.9±0.62 | 35.6±0.28 |
| | | 5 | 9.3±0.80 | 19.5±1.61 | 21.0±1.67 | 7.1±0.31 | 4.1±0.68 | 7.4±0.49 | 12.6±1.07 | 14.6±0.31 | 16.7±1.29 | 17.9±1.72 | 26.1±0.30 | 24.3±1.79 | 32.5±1.43 | 15.5±0.84 |
| | Elastic Trans. | 1 | 58.9±0.39 | 75.2±0.44 | 75.6±0.41 | 62.1±0.35 | 69.1±3.10 | 75.2±0.14 | 76.3±0.14 | 76.9±0.30 | 77.9±0.15 | 78.3±0.47 | 79.1±0.31 | 79.0±0.15 | 79.4±0.09 | 76.2±0.62 |
| | | 2 | 52.3±0.16 | 67.2±0.21 | 67.0±0.11 | 53.8±0.43 | 59.2±2.77 | 65.5±0.46 | 66.4±0.21 | 67.0±0.18 | 68.4±0.02 | 69.0±0.28 | 69.1±0.19 | 69.2±0.06 | 69.2±0.28 | 66.2±0.55 |
| | | 3 | 58.3±0.27 | 74.7±0.53 | 74.7±0.21 | 62.2±0.21 | 68.8±3.37 | 75.0±0.30 | 76.0±0.68 | 76.2±0.12 | 77.2±0.48 | 77.5±0.22 | 77.9±0.39 | 78.1±0.19 | 78.4±0.22 | 75.7±0.76 |
| | | 4 | 56.6±0.46 | 72.1±0.48 | 71.8±0.66 | 60.4±0.22 | 66.9±3.95 | 72.7±0.36 | 74.1±0.25 | 74.0±0.39 | 75.3±0.20 | 75.3±0.24 | 75.5±0.24 | 75.7±0.12 | 76.0±0.19 | 73.4±0.51 |
| | | 5 | 50.0±0.54 | 62.5±0.36 | 61.6±0.50 | 52.2±0.34 | 57.3±4.70 | 63.0±0.52 | 63.7±0.19 | 64.7±0.93 | 65.2±0.36 | 64.8±0.35 | 64.7±0.17 | 65.1±0.26 | 65.4±0.43 | 63.8±0.50 |
| | JPEG Comp. | 1 | 52.7±0.83 | 67.7±0.36 | 68.7±0.35 | 58.6±0.39 | 66.9±1.86 | 70.9±0.24 | 72.1±0.64 | 71.7±0.42 | 72.8±0.15 | 72.8±0.48 | 73.5±0.48 | 73.8±0.34 | 74.2±0.27 | 70.8±1.19 |
| | | 2 | 48.7±1.18 | 63.9±0.54 | 64.0±0.09 | 55.4±0.57 | 64.7±1.90 | 67.9±0.69 | 68.8±0.62 | 68.8±0.10 | 69.8±0.60 | 69.2±0.44 | 69.8±0.57 | 69.8±0.57 | 71.2±0.81 | 67.6±0.83 |
| | | 3 | 45.7±1.55 | 60.4±0.29 | 60.8±0.15 | 53.1±0.22 | 62.3±2.10 | 65.8±0.52 | 66.8±0.48 | 66.3±0.40 | 67.1±0.54 | 66.5±0.51 | 67.1±0.68 | 67.6±0.55 | 68.7±0.19 | 65.1±0.68 |
| | | 4 | 36.7±1.05 | 51.1±0.32 | 50.7±0.32 | 46.5±0.66 | 56.3±1.06 | 59.0±0.94 | 59.2±0.61 | 58.3±0.35 | 58.5±0.83 | 58.0±0.17 | 58.1±0.64 | 58.9±1.09 | 60.9±0.27 | 56.9±1.21 |
| | | 5 | 28.4±1.40 | 39.1±0.58 | 38.5±0.39 | 38.3±0.68 | 47.9±1.01 | 49.2±1.20 | 48.8±0.43 | 47.1±0.44 | 46.7±1.42 | 45.9±0.23 | 46.2±0.63 | 46.8±0.94 | 49.1±0.35 | 45.4±1.06 |
| | Pixelate | 1 | 60.6±0.49 | 77.3±0.33 | 77.2±0.27 | 64.4±0.25 | 71.8±2.93 | 77.1±0.17 | 78.4±0.07 | 78.8±0.38 | 79.7±0.20 | 80.0±0.18 | 80.7±0.43 | 80.8±0.13 | 81.2±0.31 | 77.3±0.35 |
| | | 2 | 59.6±0.52 | 76.3±0.32 | 76.1±0.46 | 63.5±0.19 | 70.6±3.40 | 76.3±0.20 | 77.8±0.25 | 78.1±0.13 | 79.2±0.11 | 79.2±0.23 | 80.0±0.29 | 80.2±0.29 | 80.5±0.09 | 76.4±0.21 |
| | | 3 | 56.0±0.19 | 71.5±0.71 | 68.8±0.17 | 59.7±0.39 | 66.0±3.17 | 72.4±0.19 | 72.9±0.63 | 73.7±0.17 | 74.4±0.15 | 74.6±0.21 | 74.4±0.49 | 75.4±0.83 | 76.3±0.23 | 68.3±0.47 |

Table 7: Comparison of robustness against natural corruptions at five different severities for FDT and DeiT models trained on ImageNet-100 dataset. Models are evaluated using the ImageNet100-O dataset. The models are labeled with their corresponding fixation budget hyperparameter (subscript) and model size (T, S, B for tiny, small, and base, respectively). The mean and one standard deviation of three runs with different initializations are reported.

| Corruption | | S | DeiT-T | DeiT-S | DeiT-B | $FDT\text{-}T_{0.5}$ | $FDT\text{-}S_{0.2}$ | $FDT\text{-}S_{0.3}$ | $FDT\text{-}S_{0.4}$ | $FDT\text{-}S_{0.5}$ | $FDT\text{-}S_{0.6}$ | $FDT\text{-}S_{0.7}$ | $FDT\text{-}S_{0.8}$ | $FDT\text{-}S_{0.9}$ | $FDT\text{-}S_{1.0}$ | $FDT\text{-}B_{0.5}$ |
|---|---|---|---|---|---|---|---|---|---|---|---|---|---|---|---|---|
| | | 4 | 50.8±0.72 | 59.9±1.42 | 55.3±1.89 | 53.4±0.10 | 58.3±5.28 | 64.4±0.68 | 63.5±0.77 | 63.8±0.35 | 62.8±0.38 | 62.0±0.29 | 61.8±0.38 | 61.5±0.95 | 62.6±1.51 | 53.7±0.96 |
| | | 5 | 45.0±0.75 | 48.6±3.35 | 45.1±2.48 | 46.7±0.44 | 49.2±4.55 | 55.8±0.61 | 53.3±0.71 | 52.3±0.62 | 51.2±1.10 | 51.5±0.51 | 49.1±0.84 | 49.2±1.10 | 51.1±2.23 | 44.9±1.27 |
| Noise | Gaussian Noise | 1 | 56.5±0.18 | 74.3±0.18 | 75.4±0.10 | 60.5±0.30 | 69.1±3.27 | 74.0±0.10 | 76.2±0.46 | 76.1±0.38 | 77.4±0.46 | 77.9±0.40 | 78.8±0.25 | 79.1±0.17 | 79.9±0.11 | 76.2±0.19 |
| | | 2 | 50.3±0.29 | 69.1±0.05 | 70.6±0.04 | 54.5±0.34 | 63.5±2.67 | 68.4±0.19 | 70.7±0.37 | 70.9±0.23 | 72.7±0.54 | 73.7±0.49 | 74.2±0.25 | 75.4±0.21 | 76.3±0.58 | 71.7±0.27 |
| | | 3 | 39.4±0.22 | 57.2±0.52 | 60.1±0.50 | 44.1±0.44 | 53.2±2.12 | 57.6±0.90 | 59.3±0.57 | 60.3±0.67 | 62.5±1.37 | 64.0±0.49 | 64.3±0.53 | 66.5±0.13 | 68.3±0.40 | 60.9±0.74 |
| | | 4 | 25.0±0.38 | 38.3±0.59 | 42.4±0.79 | 30.4±1.33 | 36.3±1.33 | 39.5±1.57 | 40.6±1.57 | 42.0±1.95 | 45.6±1.30 | 46.4±1.19 | 46.8±0.65 | 50.8±1.33 | 54.0±1.09 | 42.8±0.84 |
| | | 5 | 10.1±0.45 | 16.1±0.41 | 18.9±0.40 | 13.4±2.24 | 15.3±1.79 | 17.0±1.74 | 17.3±1.35 | 18.2±2.09 | 21.7±0.67 | 21.3±1.25 | 22.6±1.03 | 26.9±2.23 | 31.4±1.01 | 18.4±0.74 |
| | Impulse Noise | 1 | 54.1±0.19 | 71.8±0.05 | 73.1±0.26 | 58.1±0.09 | 67.6±2.63 | 72.1±0.19 | 73.8±0.35 | 73.9±0.31 | 75.5±0.29 | 75.8±0.71 | 76.6±0.55 | 77.6±0.28 | 78.1±0.40 | 74.2±0.50 |
| | | 2 | 45.5±0.21 | 63.8±0.26 | 66.3±0.21 | 50.2±0.08 | 60.2±2.01 | 64.1±0.35 | 66.7±0.22 | 66.9±0.59 | 68.9±0.95 | 69.9±0.72 | 70.4±0.87 | 72.1±0.33 | 72.8±0.32 | 67.2±0.61 |
| | | 3 | 38.1±0.10 | 55.8±0.59 | 58.9±0.62 | 43.2±0.65 | 52.6±1.50 | 56.9±0.90 | 59.1±0.89 | 59.2±1.40 | 61.8±1.46 | 63.1±0.79 | 63.0±0.59 | 65.8±0.11 | 66.7±0.48 | 59.7±0.49 |
| | | 4 | 22.5±0.53 | 35.0±0.63 | 39.4±1.14 | 27.3±1.24 | 33.0±1.95 | 36.9±1.26 | 38.6±1.23 | 39.0±1.93 | 43.1±1.03 | 43.6±1.29 | 43.9±0.62 | 48.7±1.76 | 51.1±1.14 | 40.1±1.18 |
| | | 5 | 10.8±0.78 | 15.8±0.36 | 19.2±0.28 | 13.1±1.73 | 15.0±2.72 | 17.1±1.97 | 17.6±1.50 | 18.0±2.06 | 22.3±0.50 | 21.4±0.99 | 23.2±1.02 | 27.9±2.47 | 32.0±1.12 | 18.6±0.78 |
| | Shot Noise | 1 | 55.9±0.21 | 74.2±0.22 | 75.3±0.26 | 60.3±0.11 | 69.0±3.48 | 73.7±0.41 | 75.6±0.32 | 76.1±0.21 | 77.2±0.13 | 77.8±0.13 | 78.4±0.27 | 79.0±0.17 | 79.9±0.14 | 76.2±0.43 |
| | | 2 | 48.3±0.44 | 67.6±0.32 | 69.1±0.17 | 53.2±0.29 | 62.7±3.09 | 67.2±0.24 | 69.3±0.19 | 69.7±0.36 | 71.7±0.76 | 72.8±0.32 | 73.1±0.79 | 74.2±0.38 | 75.3±0.38 | 70.5±0.16 |
| | | 3 | 38.0±0.19 | 55.4±0.54 | 57.9±1.19 | 43.7±0.22 | 52.2±2.37 | 56.1±0.84 | 58.5±0.40 | 58.8±0.91 | 61.7±1.37 | 63.2±0.84 | 63.2±0.78 | 65.1±0.25 | 67.0±0.40 | 59.0±0.73 |
| | | 4 | 21.8±0.33 | 32.7±0.60 | 35.8±0.90 | 26.8±1.12 | 32.3±1.38 | 34.6±1.87 | 35.5±1.58 | 36.4±1.92 | 40.4±1.03 | 40.6±1.36 | 41.3±1.02 | 45.3±1.62 | 48.7±1.08 | 35.8±1.01 |
| | | 5 | 12.7±0.44 | 18.4±0.75 | 20.8±0.69 | 16.1±1.59 | 18.8±1.35 | 20.0±1.18 | 20.2±1.68 | 21.1±2.00 | 25.3±0.98 | 24.9±1.34 | 25.9±1.42 | 30.1±2.02 | 33.4±1.23 | 20.0±0.99 |
| Weather | Brightness | 1 | 62.4±0.28 | 78.8±0.09 | 79.1±0.27 | 65.9±0.19 | 73.4±2.32 | 78.2±0.24 | 79.5±0.51 | 80.1±0.32 | 80.7±0.11 | 81.4±0.24 | 81.9±0.25 | 81.9±0.33 | 82.0±0.08 | 80.1±0.12 |
| | | 2 | 60.6±0.33 | 77.2±0.14 | 78.0±0.26 | 64.0±0.35 | 72.2±1.80 | 76.6±0.07 | 78.6±0.46 | 78.7±0.07 | 79.6±0.29 | 80.3±0.24 | 80.8±0.30 | 81.1±0.30 | 81.2±0.05 | 78.8±0.35 |
| | | 3 | 58.0±0.30 | 75.2±0.30 | 76.0±0.13 | 61.1±0.33 | 69.6±1.85 | 74.4±0.12 | 76.5±0.14 | 77.3±0.38 | 78.0±0.25 | 78.4±0.29 | 79.0±0.07 | 79.4±0.10 | 79.7±0.10 | 77.0±0.22 |
| | | 4 | 53.0±0.30 | 71.3±0.18 | 73.1±0.38 | 57.0±0.48 | 65.9±1.22 | 70.6±0.22 | 73.5±0.21 | 74.1±0.43 | 75.2±0.24 | 75.6±0.14 | 76.6±0.19 | 76.6±0.35 | 77.2±0.37 | 74.5±0.22 |
| | | 5 | 46.7±0.40 | 65.5±0.21 | 67.6±0.40 | 50.5±0.33 | 59.9±1.06 | 64.9±0.14 | 68.5±0.20 | 69.3±0.11 | 70.6±0.18 | 70.9±0.17 | 71.7±0.32 | 72.7±0.62 | 73.1±0.27 | 69.9±0.21 |
| | Fog | 1 | 48.3±1.08 | 68.0±0.19 | 69.4±0.60 | 52.1±0.44 | 59.7±3.21 | 67.7±0.34 | 69.8±0.49 | 70.2±0.21 | 71.2±0.69 | 71.9±0.90 | 73.3±0.54 | 73.3±0.41 | 73.3±0.16 | 70.8±0.43 |
| | | 2 | 43.0±0.86 | 62.4±0.44 | 63.7±0.85 | 45.6±0.54 | 53.8±2.42 | 61.9±0.34 | 64.2±0.28 | 64.8±0.48 | 65.0±1.31 | 66.4±1.14 | 67.4±0.72 | 67.6±0.89 | 68.3±0.39 | 65.6±0.37 |
| | | 3 | 35.4±0.73 | 53.5±0.35 | 55.0±1.02 | 37.2±1.06 | 45.3±2.31 | 53.5±0.56 | 55.8±0.16 | 56.0±0.45 | 55.3±1.28 | 56.9±1.78 | 58.0±1.14 | 58.3±0.78 | 59.1±0.67 | 56.8±0.96 |
| | | 4 | 32.0±0.73 | 49.4±1.15 | 51.5±0.68 | 34.0±0.84 | 42.5±2.44 | 49.8±0.83 | 52.1±0.19 | 52.4±0.24 | 51.7±1.17 | 53.3±1.81 | 54.7±0.95 | 54.7±0.67 | 56.0±0.45 | 52.2±0.74 |
| | | 5 | 24.1±0.80 | 38.3±1.34 | 41.8±0.95 | 25.7±0.29 | 32.7±2.09 | 38.5±0.65 | 41.0±1.05 | 43.1±0.48 | 42.4±1.35 | 44.6±1.42 | 45.7±1.06 | 46.6±0.46 | 47.4±0.36 | 41.3±1.46 |
| | Frost | 1 | 55.9±0.10 | 73.2±0.36 | 74.6±0.58 | 59.5±0.19 | 69.1±2.48 | 73.4±0.22 | 74.8±0.66 | 75.8±0.19 | 77.2±0.18 | 77.2±0.51 | 77.7±0.29 | 78.0±0.18 | 78.3±0.29 | 75.6±0.75 |
| | | 2 | 46.4±1.02 | 66.5±0.37 | 68.1±0.52 | 50.2±0.31 | 61.4±2.78 | 65.2±0.12 | 67.9±0.66 | 69.3±0.28 | 70.8±0.25 | 70.7±0.21 | 71.5±0.32 | 72.3±0.05 | 72.8±0.04 | 69.6±0.60 |
| | | 3 | 39.0±0.76 | 59.8±0.84 | 61.9±0.71 | 42.7±0.61 | 54.3±2.39 | 58.3±0.24 | 61.4±0.56 | 63.2±0.52 | 64.3±0.21 | 64.5±0.26 | 65.7±0.47 | 66.5±0.70 | 67.5±0.25 | 63.6±0.95 |
| | | 4 | 38.6±0.98 | 59.2±0.85 | 61.3±0.68 | 41.5±0.19 | 53.3±2.46 | 57.2±0.28 | 60.2±0.63 | 62.2±0.56 | 63.1±0.13 | 63.4±0.06 | 64.6±0.57 | 65.7±0.46 | 66.6±0.19 | 62.5±1.28 |
| | | 5 | 34.2±1.26 | 54.1±0.60 | 56.1±0.48 | 36.7±0.32 | 48.1±2.71 | 52.1±0.53 | 55.6±0.71 | 57.0±0.78 | 58.2±0.36 | 58.8±0.26 | 59.9±0.57 | 61.1±0.91 | 61.9±0.35 | 57.9±1.07 |

19

Table 7: Comparison of robustness against natural corruptions at five different severities for FDT and DeiT models trained on ImageNet-100 dataset. Models are evaluated using the ImageNet100-O dataset. The models are labeled with their corresponding fixation budget hyperparameter (subscript) and model size (T, S, B for tiny, small, and base, respectively). The mean and one standard deviation of three runs with different initializations are reported.

| Corruption | S | DeiT-T | DeiT-S | DeiT-B | FDT-T$_{0.5}$ | FDT-S$_{0.2}$ | FDT-S$_{0.3}$ | FDT-S$_{0.4}$ | FDT-S$_{0.5}$ | FDT-S$_{0.6}$ | FDT-S$_{0.7}$ | FDT-S$_{0.8}$ | FDT-S$_{0.9}$ | FDT-S$_{1.0}$ | FDT-B$_{0.5}$ |
|---|---|---|---|---|---|---|---|---|---|---|---|---|---|---|---|
| Snow | 1 | $49.3_{\pm0.29}$ | $66.9_{\pm0.56}$ | $68.1_{\pm0.59}$ | $53.5_{\pm0.28}$ | $59.8_{\pm5.07}$ | $66.1_{\pm0.22}$ | $68.1_{\pm0.73}$ | $69.0_{\pm0.24}$ | $70.7_{\pm0.10}$ | $70.9_{\pm0.33}$ | $72.2_{\pm0.55}$ | $72.8_{\pm0.26}$ | $73.4_{\pm0.08}$ | $70.1_{\pm0.38}$ |
|  | 2 | $35.9_{\pm0.27}$ | $52.0_{\pm0.28}$ | $53.5_{\pm0.88}$ | $39.8_{\pm0.39}$ | $43.8_{\pm5.09}$ | $51.5_{\pm0.23}$ | $54.6_{\pm0.67}$ | $55.4_{\pm0.46}$ | $58.1_{\pm0.69}$ | $58.4_{\pm0.32}$ | $60.0_{\pm1.00}$ | $61.1_{\pm0.72}$ | $62.1_{\pm0.29}$ | $57.3_{\pm0.58}$ |
|  | 3 | $37.3_{\pm0.83}$ | $53.7_{\pm0.42}$ | $56.0_{\pm0.34}$ | $41.0_{\pm0.74}$ | $46.7_{\pm4.77}$ | $53.4_{\pm0.31}$ | $56.2_{\pm1.18}$ | $57.9_{\pm0.81}$ | $59.9_{\pm0.42}$ | $60.6_{\pm0.34}$ | $62.3_{\pm0.74}$ | $63.0_{\pm0.45}$ | $64.0_{\pm0.70}$ | $59.3_{\pm0.49}$ |
|  | 4 | $29.7_{\pm0.68}$ | $44.7_{\pm0.22}$ | $46.5_{\pm0.84}$ | $32.4_{\pm0.50}$ | $36.4_{\pm4.13}$ | $43.3_{\pm0.63}$ | $46.6_{\pm1.34}$ | $48.8_{\pm0.68}$ | $50.5_{\pm0.75}$ | $51.0_{\pm0.49}$ | $53.6_{\pm1.00}$ | $54.7_{\pm0.58}$ | $56.0_{\pm0.35}$ | $50.9_{\pm1.06}$ |
|  | 5 | $27.0_{\pm0.47}$ | $41.9_{\pm0.53}$ | $43.2_{\pm1.12}$ | $29.8_{\pm0.77}$ | $33.6_{\pm3.59}$ | $40.5_{\pm0.82}$ | $44.6_{\pm1.34}$ | $46.6_{\pm0.12}$ | $48.3_{\pm0.82}$ | $48.4_{\pm0.47}$ | $50.7_{\pm0.97}$ | $51.2_{\pm0.94}$ | $53.0_{\pm0.23}$ | $48.0_{\pm1.44}$ |

Table 8: Comparison of robustness against natural corruptions at five different severities for FDT and DeiT models trained on Tinted-ImageNet-100 dataset. The models are evaluated using the ImageNet100-O dataset. The models are labeled with their corresponding fixation budget hyperparameter (subscript) and model size (T, S, B for tiny, small, and base, respectively). The mean and one standard deviation of three runs with different initializations are reported.

| Corruption | | S | DeiT-S | FDT-S$_{0.2}$ | FDT-S$_{0.3}$ | FDT-S$_{0.4}$ | FDT-S$_{0.5}$ | FDT-S$_{0.6}$ | FDT-S$_{0.7}$ | FDT-S$_{0.8}$ | FDT-S$_{0.9}$ | FDT-S$_{1.0}$ |
|---|---|---|---|---|---|---|---|---|---|---|---|---|
| – | | 0 | $53.5_{\pm0.16}$ | $44.8_{\pm1.07}$ | $51.4_{\pm0.56}$ | $54.0_{\pm0.29}$ | $56.5_{\pm0.53}$ | $57.9_{\pm0.29}$ | $59.5_{\pm0.19}$ | $61.1_{\pm0.34}$ | $61.8_{\pm0.72}$ | $62.4_{\pm0.18}$ |
| Blur | Defocus Blur | 1 | $37.7_{\pm0.67}$ | $25.7_{\pm1.12}$ | $31.7_{\pm0.73}$ | $36.1_{\pm0.77}$ | $38.9_{\pm0.69}$ | $41.4_{\pm1.73}$ | $42.7_{\pm0.15}$ | $44.6_{\pm0.39}$ | $44.8_{\pm0.71}$ | $45.9_{\pm0.38}$ |
|  |  | 2 | $31.1_{\pm1.01}$ | $18.9_{\pm1.24}$ | $24.9_{\pm1.04}$ | $29.4_{\pm1.35}$ | $31.6_{\pm1.02}$ | $34.2_{\pm1.85}$ | $35.9_{\pm0.33}$ | $37.7_{\pm0.50}$ | $38.0_{\pm0.47}$ | $38.9_{\pm0.39}$ |
|  |  | 3 | $21.2_{\pm0.80}$ | $10.4_{\pm1.56}$ | $15.7_{\pm0.81}$ | $18.7_{\pm0.26}$ | $21.0_{\pm1.37}$ | $22.4_{\pm1.75}$ | $24.4_{\pm0.91}$ | $25.9_{\pm0.35}$ | $26.0_{\pm0.19}$ | $26.6_{\pm0.75}$ |
|  |  | 4 | $15.4_{\pm0.70}$ | $6.6_{\pm1.36}$ | $10.6_{\pm0.57}$ | $13.0_{\pm0.26}$ | $14.4_{\pm0.85}$ | $15.4_{\pm1.20}$ | $16.7_{\pm0.63}$ | $18.1_{\pm0.49}$ | $18.1_{\pm0.28}$ | $18.6_{\pm0.70}$ |
|  |  | 5 | $11.7_{\pm0.52}$ | $4.6_{\pm1.00}$ | $8.2_{\pm0.31}$ | $9.5_{\pm0.15}$ | $10.2_{\pm0.49}$ | $11.1_{\pm0.90}$ | $12.0_{\pm0.42}$ | $12.9_{\pm0.24}$ | $12.9_{\pm0.24}$ | $12.9_{\pm0.53}$ |
|  | Glass Blur | 1 | $43.7_{\pm0.16}$ | $33.5_{\pm1.37}$ | $40.0_{\pm1.04}$ | $43.4_{\pm0.68}$ | $46.0_{\pm0.21}$ | $47.8_{\pm0.88}$ | $49.4_{\pm0.62}$ | $50.9_{\pm0.82}$ | $51.0_{\pm1.03}$ | $51.9_{\pm0.19}$ |
|  |  | 2 | $36.6_{\pm0.46}$ | $26.9_{\pm1.58}$ | $33.0_{\pm0.83}$ | $37.0_{\pm0.56}$ | $38.9_{\pm0.62}$ | $40.8_{\pm1.57}$ | $42.0_{\pm0.55}$ | $43.7_{\pm0.92}$ | $43.7_{\pm0.90}$ | $44.1_{\pm0.34}$ |
|  |  | 3 | $27.2_{\pm0.25}$ | $20.6_{\pm1.16}$ | $25.8_{\pm0.92}$ | $28.7_{\pm0.16}$ | $29.7_{\pm1.17}$ | $30.6_{\pm1.74}$ | $31.0_{\pm0.54}$ | $31.4_{\pm0.86}$ | $32.1_{\pm1.34}$ | $32.2_{\pm0.54}$ |
|  |  | 4 | $22.6_{\pm0.20}$ | $15.9_{\pm0.92}$ | $20.5_{\pm0.65}$ | $23.4_{\pm0.10}$ | $24.1_{\pm1.38}$ | $24.6_{\pm1.66}$ | $25.2_{\pm0.89}$ | $25.5_{\pm0.98}$ | $26.0_{\pm0.99}$ | $26.3_{\pm0.45}$ |
|  |  | 5 | $17.0_{\pm0.18}$ | $9.6_{\pm0.92}$ | $13.6_{\pm0.25}$ | $16.0_{\pm0.17}$ | $17.0_{\pm1.10}$ | $18.0_{\pm1.49}$ | $18.7_{\pm0.43}$ | $19.1_{\pm0.61}$ | $19.2_{\pm0.38}$ | $19.4_{\pm0.14}$ |
|  | Motion Blur | 1 | $44.2_{\pm0.06}$ | $34.1_{\pm1.92}$ | $41.1_{\pm0.77}$ | $44.6_{\pm0.53}$ | $46.0_{\pm0.21}$ | $48.2_{\pm0.63}$ | $49.9_{\pm0.80}$ | $50.9_{\pm0.52}$ | $51.5_{\pm0.93}$ | $52.3_{\pm0.30}$ |
|  |  | 2 | $37.5_{\pm0.18}$ | $26.9_{\pm1.86}$ | $34.0_{\pm0.74}$ | $37.6_{\pm0.18}$ | $38.6_{\pm0.35}$ | $40.8_{\pm0.89}$ | $42.3_{\pm0.36}$ | $43.1_{\pm0.24}$ | $44.0_{\pm0.19}$ | $44.7_{\pm0.39}$ |
|  |  | 3 | $29.6_{\pm0.28}$ | $19.1_{\pm1.83}$ | $25.6_{\pm1.02}$ | $29.2_{\pm0.18}$ | $30.0_{\pm0.58}$ | $31.7_{\pm1.34}$ | $33.6_{\pm0.36}$ | $33.7_{\pm0.02}$ | $34.0_{\pm0.26}$ | $34.8_{\pm0.05}$ |
|  |  | 4 | $22.2_{\pm0.23}$ | $13.2_{\pm2.06}$ | $18.7_{\pm0.79}$ | $21.6_{\pm0.23}$ | $22.3_{\pm1.23}$ | $23.5_{\pm1.31}$ | $24.8_{\pm0.72}$ | $24.9_{\pm0.18}$ | $25.4_{\pm0.10}$ | $25.8_{\pm0.54}$ |
|  |  | 5 | $18.6_{\pm0.17}$ | $10.6_{\pm1.46}$ | $15.2_{\pm0.41}$ | $17.7_{\pm0.40}$ | $18.4_{\pm1.20}$ | $19.3_{\pm1.23}$ | $20.5_{\pm0.66}$ | $20.4_{\pm0.13}$ | $20.8_{\pm0.38}$ | $21.4_{\pm0.11}$ |

20

Table 8: Comparison of robustness against natural corruptions at five different severities for FDT and DeiT models trained on Tinted-ImageNet-100 dataset. The models are evaluated using the ImageNet100-O dataset. The models are labeled with their corresponding fixation budget hyperparameter (subscript) and model size (T, S, B for tiny, small, and base, respectively). The mean and one standard deviation of three runs with different initializations are reported.

| Corruption | | S | DeiT-S | FDT-S$_{0.2}$ | FDT-S$_{0.3}$ | FDT-S$_{0.4}$ | FDT-S$_{0.5}$ | FDT-S$_{0.6}$ | FDT-S$_{0.7}$ | FDT-S$_{0.8}$ | FDT-S$_{0.9}$ | FDT-S$_{1.0}$ |
|---|---|---|---|---|---|---|---|---|---|---|---|---|
| | Zoom Blur | 1 | $39.8_{\pm0.29}$ | $30.9_{\pm1.34}$ | $37.6_{\pm1.00}$ | $40.0_{\pm0.22}$ | $41.6_{\pm0.52}$ | $42.5_{\pm0.96}$ | $42.9_{\pm0.32}$ | $43.3_{\pm0.24}$ | $43.8_{\pm0.51}$ | $43.8_{\pm0.34}$ |
| | | 2 | $34.8_{\pm0.42}$ | $26.8_{\pm1.40}$ | $32.6_{\pm1.26}$ | $34.6_{\pm0.16}$ | $36.7_{\pm0.43}$ | $37.1_{\pm1.35}$ | $37.1_{\pm0.39}$ | $37.6_{\pm0.20}$ | $38.4_{\pm0.51}$ | $38.3_{\pm0.11}$ |
| | | 3 | $32.2_{\pm0.47}$ | $23.2_{\pm1.57}$ | $29.1_{\pm1.06}$ | $31.3_{\pm0.27}$ | $33.3_{\pm0.57}$ | $33.9_{\pm1.48}$ | $34.0_{\pm0.75}$ | $34.2_{\pm0.42}$ | $34.6_{\pm0.35}$ | $34.8_{\pm0.42}$ |
| | | 4 | $29.2_{\pm0.32}$ | $20.8_{\pm1.32}$ | $26.4_{\pm1.02}$ | $28.5_{\pm0.66}$ | $30.2_{\pm0.61}$ | $30.5_{\pm1.39}$ | $30.8_{\pm0.44}$ | $30.9_{\pm0.59}$ | $31.1_{\pm0.44}$ | $31.4_{\pm0.10}$ |
| | | 5 | $26.0_{\pm0.62}$ | $17.7_{\pm1.34}$ | $23.6_{\pm1.07}$ | $25.1_{\pm0.61}$ | $26.6_{\pm0.48}$ | $27.4_{\pm1.45}$ | $27.1_{\pm0.46}$ | $27.4_{\pm0.06}$ | $27.4_{\pm0.40}$ | $28.2_{\pm0.23}$ |
| Digital | Contrast | 1 | $44.1_{\pm0.37}$ | $27.2_{\pm0.48}$ | $37.5_{\pm0.71}$ | $42.2_{\pm1.03}$ | $44.9_{\pm1.03}$ | $47.7_{\pm0.88}$ | $49.8_{\pm0.65}$ | $51.6_{\pm0.61}$ | $52.5_{\pm0.37}$ | $53.2_{\pm0.18}$ |
| | | 2 | $40.6_{\pm0.27}$ | $21.6_{\pm0.14}$ | $32.6_{\pm0.56}$ | $37.6_{\pm0.85}$ | $41.1_{\pm1.24}$ | $43.7_{\pm0.59}$ | $46.8_{\pm0.93}$ | $47.6_{\pm0.08}$ | $48.9_{\pm0.46}$ | $49.4_{\pm0.28}$ |
| | | 3 | $34.4_{\pm0.89}$ | $14.3_{\pm0.23}$ | $24.8_{\pm0.57}$ | $30.5_{\pm0.49}$ | $35.2_{\pm1.33}$ | $38.0_{\pm0.76}$ | $41.5_{\pm0.99}$ | $42.1_{\pm0.46}$ | $43.4_{\pm1.08}$ | $43.9_{\pm0.51}$ |
| | | 4 | $20.4_{\pm0.60}$ | $6.4_{\pm0.39}$ | $12.6_{\pm0.70}$ | $15.9_{\pm0.44}$ | $20.9_{\pm1.33}$ | $24.6_{\pm1.61}$ | $26.8_{\pm1.18}$ | $28.2_{\pm0.89}$ | $28.7_{\pm1.78}$ | $27.6_{\pm0.77}$ |
| | | 5 | $5.9_{\pm0.51}$ | $2.4_{\pm0.13}$ | $4.1_{\pm0.55}$ | $5.1_{\pm0.43}$ | $7.4_{\pm1.05}$ | $8.6_{\pm0.18}$ | $8.3_{\pm0.59}$ | $8.9_{\pm0.46}$ | $8.7_{\pm1.04}$ | $8.1_{\pm0.72}$ |
| | Elastic Trans. | 1 | $48.7_{\pm0.23}$ | $39.9_{\pm1.64}$ | $46.5_{\pm0.95}$ | $49.1_{\pm0.56}$ | $51.4_{\pm0.34}$ | $52.8_{\pm0.20}$ | $53.6_{\pm0.09}$ | $55.3_{\pm0.34}$ | $56.0_{\pm1.16}$ | $56.2_{\pm0.36}$ |
| | | 2 | $42.1_{\pm0.33}$ | $32.3_{\pm1.55}$ | $38.6_{\pm0.47}$ | $41.3_{\pm0.70}$ | $42.5_{\pm0.03}$ | $44.1_{\pm0.26}$ | $45.0_{\pm0.16}$ | $46.0_{\pm0.36}$ | $46.5_{\pm0.85}$ | $47.0_{\pm0.07}$ |
| | | 3 | $48.6_{\pm0.43}$ | $40.4_{\pm1.53}$ | $47.2_{\pm0.74}$ | $49.5_{\pm0.38}$ | $51.9_{\pm0.10}$ | $53.2_{\pm0.42}$ | $54.5_{\pm0.70}$ | $56.4_{\pm0.47}$ | $56.5_{\pm1.18}$ | $56.6_{\pm0.49}$ |
| | | 4 | $46.2_{\pm0.35}$ | $39.1_{\pm1.35}$ | $45.4_{\pm0.99}$ | $47.7_{\pm0.24}$ | $50.0_{\pm0.44}$ | $51.2_{\pm0.37}$ | $52.0_{\pm0.45}$ | $53.7_{\pm0.65}$ | $54.0_{\pm1.21}$ | $54.1_{\pm0.50}$ |
| | | 5 | $38.7_{\pm0.57}$ | $33.7_{\pm0.91}$ | $39.1_{\pm0.97}$ | $40.7_{\pm0.22}$ | $42.9_{\pm0.37}$ | $43.4_{\pm0.48}$ | $43.8_{\pm0.42}$ | $45.3_{\pm0.63}$ | $45.3_{\pm1.30}$ | $45.5_{\pm0.75}$ |
| | JPEG Comp. | 1 | $45.3_{\pm0.24}$ | $42.0_{\pm1.39}$ | $47.5_{\pm0.19}$ | $48.8_{\pm0.36}$ | $51.0_{\pm0.29}$ | $52.0_{\pm0.30}$ | $52.4_{\pm0.29}$ | $54.1_{\pm0.65}$ | $54.2_{\pm1.34}$ | $54.7_{\pm0.33}$ |
| | | 2 | $41.4_{\pm0.29}$ | $40.1_{\pm1.39}$ | $45.1_{\pm0.19}$ | $46.3_{\pm0.52}$ | $47.8_{\pm0.15}$ | $48.6_{\pm0.42}$ | $48.7_{\pm0.57}$ | $50.4_{\pm0.50}$ | $50.0_{\pm1.00}$ | $51.3_{\pm0.15}$ |
| | | 3 | $38.3_{\pm0.46}$ | $37.8_{\pm1.31}$ | $43.0_{\pm0.40}$ | $43.3_{\pm0.12}$ | $45.1_{\pm0.21}$ | $45.9_{\pm0.50}$ | $46.1_{\pm0.65}$ | $47.7_{\pm0.25}$ | $47.6_{\pm0.43}$ | $48.3_{\pm0.28}$ |
| | | 4 | $28.5_{\pm0.29}$ | $32.4_{\pm1.36}$ | $36.0_{\pm0.32}$ | $35.2_{\pm0.52}$ | $36.8_{\pm0.30}$ | $36.9_{\pm0.45}$ | $37.0_{\pm0.71}$ | $38.5_{\pm0.68}$ | $38.7_{\pm0.33}$ | $38.8_{\pm0.29}$ |
| | | 5 | $17.7_{\pm0.47}$ | $26.1_{\pm1.84}$ | $27.4_{\pm0.61}$ | $25.9_{\pm1.12}$ | $27.5_{\pm0.22}$ | $26.4_{\pm1.40}$ | $26.2_{\pm1.35}$ | $27.8_{\pm0.94}$ | $27.6_{\pm0.21}$ | $27.6_{\pm0.99}$ |
| | Pixelate | 1 | $52.6_{\pm0.20}$ | $42.3_{\pm1.24}$ | $49.1_{\pm0.96}$ | $52.3_{\pm0.26}$ | $54.0_{\pm0.31}$ | $55.9_{\pm0.33}$ | $57.4_{\pm0.47}$ | $59.2_{\pm0.36}$ | $59.6_{\pm0.79}$ | $60.8_{\pm0.21}$ |
| | | 2 | $51.4_{\pm0.61}$ | $40.8_{\pm1.62}$ | $47.9_{\pm0.44}$ | $51.2_{\pm0.23}$ | $53.3_{\pm0.08}$ | $55.0_{\pm0.68}$ | $56.7_{\pm0.61}$ | $58.4_{\pm0.63}$ | $59.1_{\pm0.78}$ | $60.0_{\pm0.16}$ |
| | | 3 | $48.6_{\pm0.08}$ | $39.4_{\pm1.53}$ | $46.9_{\pm0.96}$ | $50.1_{\pm0.51}$ | $51.6_{\pm0.25}$ | $52.7_{\pm0.47}$ | $53.5_{\pm0.32}$ | $54.6_{\pm0.31}$ | $55.6_{\pm0.45}$ | $55.9_{\pm0.57}$ |
| | | 4 | $42.0_{\pm0.22}$ | $35.3_{\pm1.55}$ | $42.1_{\pm1.36}$ | $44.1_{\pm0.67}$ | $45.5_{\pm0.89}$ | $45.8_{\pm0.79}$ | $46.1_{\pm0.12}$ | $46.5_{\pm0.59}$ | $47.7_{\pm0.49}$ | $47.4_{\pm1.04}$ |
| | | 5 | $33.4_{\pm0.84}$ | $30.7_{\pm1.69}$ | $36.0_{\pm1.68}$ | $37.3_{\pm1.46}$ | $37.4_{\pm1.43}$ | $37.6_{\pm0.64}$ | $37.9_{\pm0.41}$ | $37.2_{\pm0.94}$ | $38.5_{\pm0.62}$ | $36.8_{\pm0.48}$ |
| Noise | Gaussian Noise | 1 | $52.6_{\pm0.33}$ | $46.2_{\pm0.73}$ | $51.6_{\pm0.65}$ | $53.9_{\pm0.11}$ | $55.8_{\pm0.50}$ | $58.1_{\pm0.34}$ | $58.8_{\pm0.07}$ | $60.5_{\pm0.30}$ | $61.0_{\pm0.20}$ | $61.8_{\pm0.32}$ |
| | | 2 | $46.6_{\pm0.32}$ | $41.2_{\pm1.41}$ | $46.2_{\pm0.18}$ | $49.3_{\pm0.42}$ | $51.3_{\pm1.00}$ | $53.2_{\pm0.58}$ | $54.2_{\pm0.20}$ | $56.1_{\pm0.30}$ | $56.2_{\pm0.59}$ | $57.7_{\pm0.25}$ |
| | | 3 | $36.9_{\pm0.94}$ | $32.9_{\pm1.83}$ | $37.0_{\pm0.35}$ | $39.3_{\pm0.61}$ | $40.5_{\pm0.73}$ | $43.6_{\pm0.42}$ | $45.1_{\pm0.31}$ | $47.2_{\pm0.94}$ | $46.9_{\pm0.91}$ | $49.3_{\pm0.41}$ |
| | | 4 | $24.3_{\pm0.40}$ | $22.2_{\pm2.05}$ | $24.7_{\pm0.41}$ | $26.1_{\pm1.23}$ | $25.2_{\pm0.82}$ | $29.1_{\pm1.11}$ | $30.1_{\pm0.72}$ | $32.6_{\pm1.69}$ | $33.0_{\pm1.55}$ | $35.6_{\pm0.67}$ |
| | | 5 | $11.2_{\pm0.47}$ | $10.5_{\pm1.64}$ | $12.2_{\pm0.68}$ | $12.1_{\pm0.94}$ | $10.8_{\pm0.72}$ | $13.1_{\pm0.67}$ | $13.2_{\pm0.46}$ | $14.5_{\pm1.55}$ | $15.0_{\pm1.92}$ | $17.7_{\pm1.49}$ |
| | Impulse Noise | 1 | $49.4_{\pm0.60}$ | $44.1_{\pm1.12}$ | $49.8_{\pm0.37}$ | $51.8_{\pm0.21}$ | $53.7_{\pm0.64}$ | $55.8_{\pm0.29}$ | $56.6_{\pm0.20}$ | $58.1_{\pm0.07}$ | $58.4_{\pm0.33}$ | $59.9_{\pm0.14}$ |
| | | 2 | $42.2_{\pm0.49}$ | $37.6_{\pm1.51}$ | $42.3_{\pm0.28}$ | $44.6_{\pm0.63}$ | $46.4_{\pm0.68}$ | $49.3_{\pm0.07}$ | $50.0_{\pm0.33}$ | $52.6_{\pm0.41}$ | $51.7_{\pm0.35}$ | $54.3_{\pm0.25}$ |

Table 8: Comparison of robustness against natural corruptions at five different severities for FDT and DeiT models trained on Tinted-ImageNet-100 dataset. The models are evaluated using the ImageNet100-O dataset. The models are labeled with their corresponding fixation budget hyperparameter (subscript) and model size (T, S, B for tiny, small, and base, respectively). The mean and one standard deviation of three runs with different initializations are reported.

| Corruption | | S | DeiT-S | FDT-S$_{0.2}$ | FDT-S$_{0.3}$ | FDT-S$_{0.4}$ | FDT-S$_{0.5}$ | FDT-S$_{0.6}$ | FDT-S$_{0.7}$ | FDT-S$_{0.8}$ | FDT-S$_{0.9}$ | FDT-S$_{1.0}$ |
|---|---|---|---|---|---|---|---|---|---|---|---|---|
| | | 3 | 35.6$_{\pm0.97}$ | 31.9$_{\pm2.11}$ | 35.6$_{\pm0.30}$ | 37.8$_{\pm0.52}$ | 39.1$_{\pm0.67}$ | 43.3$_{\pm0.15}$ | 44.2$_{\pm0.12}$ | 46.5$_{\pm0.88}$ | 45.9$_{\pm0.89}$ | 48.6$_{\pm0.19}$ |
| | | 4 | 22.5$_{\pm1.16}$ | 20.0$_{\pm2.45}$ | 22.4$_{\pm0.63}$ | 23.7$_{\pm1.03}$ | 23.1$_{\pm0.67}$ | 27.2$_{\pm0.95}$ | 28.7$_{\pm0.42}$ | 30.9$_{\pm1.44}$ | 31.1$_{\pm1.48}$ | 33.7$_{\pm0.60}$ |
| | | 5 | 11.6$_{\pm1.02}$ | 10.5$_{\pm1.52}$ | 11.9$_{\pm0.47}$ | 11.7$_{\pm0.96}$ | 10.9$_{\pm0.47}$ | 13.9$_{\pm1.07}$ | 14.4$_{\pm0.58}$ | 15.2$_{\pm1.38}$ | 14.9$_{\pm1.91}$ | 18.2$_{\pm1.04}$ |
| | Shot Noise | 1 | 51.4$_{\pm0.18}$ | 44.9$_{\pm1.08}$ | 50.9$_{\pm0.82}$ | 52.9$_{\pm0.11}$ | 54.7$_{\pm0.55}$ | 56.5$_{\pm0.19}$ | 57.4$_{\pm0.11}$ | 59.5$_{\pm0.18}$ | 59.8$_{\pm0.21}$ | 61.1$_{\pm0.25}$ |
| | | 2 | 44.7$_{\pm0.62}$ | 39.6$_{\pm1.44}$ | 44.8$_{\pm0.45}$ | 47.0$_{\pm0.44}$ | 49.1$_{\pm0.86}$ | 51.0$_{\pm0.19}$ | 52.0$_{\pm0.40}$ | 54.1$_{\pm0.68}$ | 54.0$_{\pm0.20}$ | 55.8$_{\pm0.30}$ |
| | | 3 | 35.3$_{\pm0.91}$ | 31.5$_{\pm1.65}$ | 36.3$_{\pm0.28}$ | 38.2$_{\pm0.68}$ | 39.7$_{\pm0.71}$ | 42.0$_{\pm0.49}$ | 43.4$_{\pm0.52}$ | 45.4$_{\pm1.01}$ | 45.8$_{\pm0.41}$ | 47.6$_{\pm0.21}$ |
| | | 4 | 20.6$_{\pm0.58}$ | 18.3$_{\pm1.92}$ | 21.8$_{\pm0.13}$ | 22.4$_{\pm0.50}$ | 21.0$_{\pm0.76}$ | 24.1$_{\pm1.42}$ | 25.2$_{\pm0.80}$ | 27.5$_{\pm1.49}$ | 27.9$_{\pm0.99}$ | 30.0$_{\pm0.68}$ |
| | | 5 | 11.8$_{\pm0.69}$ | 11.3$_{\pm1.47}$ | 13.0$_{\pm0.25}$ | 13.3$_{\pm0.95}$ | 11.7$_{\pm0.44}$ | 14.2$_{\pm1.07}$ | 14.5$_{\pm0.49}$ | 16.1$_{\pm1.21}$ | 16.3$_{\pm1.06}$ | 18.8$_{\pm1.11}$ |
| Weather | Brightness | 1 | 48.8$_{\pm0.23}$ | 41.1$_{\pm1.06}$ | 47.2$_{\pm0.71}$ | 49.1$_{\pm0.45}$ | 51.4$_{\pm0.16}$ | 53.7$_{\pm0.37}$ | 54.7$_{\pm0.71}$ | 56.1$_{\pm0.87}$ | 56.7$_{\pm0.74}$ | 57.6$_{\pm0.78}$ |
| | | 2 | 45.6$_{\pm0.28}$ | 38.4$_{\pm0.37}$ | 44.3$_{\pm0.42}$ | 46.7$_{\pm0.74}$ | 49.0$_{\pm0.16}$ | 51.5$_{\pm0.44}$ | 52.5$_{\pm0.70}$ | 53.8$_{\pm0.80}$ | 54.3$_{\pm0.24}$ | 55.1$_{\pm0.30}$ |
| | | 3 | 44.3$_{\pm0.24}$ | 37.1$_{\pm0.36}$ | 43.0$_{\pm0.63}$ | 45.4$_{\pm0.48}$ | 47.7$_{\pm0.15}$ | 50.0$_{\pm0.21}$ | 51.1$_{\pm0.69}$ | 52.2$_{\pm0.68}$ | 52.9$_{\pm0.03}$ | 53.8$_{\pm0.51}$ |
| | | 4 | 41.5$_{\pm0.53}$ | 34.5$_{\pm0.46}$ | 40.5$_{\pm0.35}$ | 43.0$_{\pm0.41}$ | 45.5$_{\pm0.50}$ | 48.2$_{\pm0.16}$ | 49.1$_{\pm0.68}$ | 50.1$_{\pm0.47}$ | 50.8$_{\pm0.09}$ | 51.7$_{\pm0.26}$ |
| | | 5 | 37.3$_{\pm0.78}$ | 30.8$_{\pm0.48}$ | 36.9$_{\pm0.60}$ | 39.4$_{\pm0.18}$ | 41.6$_{\pm0.42}$ | 43.9$_{\pm0.28}$ | 45.1$_{\pm0.75}$ | 46.3$_{\pm0.33}$ | 46.9$_{\pm0.27}$ | 48.2$_{\pm0.37}$ |
| | Fog | 1 | 42.5$_{\pm0.44}$ | 32.1$_{\pm1.84}$ | 40.3$_{\pm0.76}$ | 41.6$_{\pm0.15}$ | 45.7$_{\pm0.25}$ | 46.8$_{\pm0.31}$ | 48.7$_{\pm0.78}$ | 49.5$_{\pm0.93}$ | 50.1$_{\pm0.49}$ | 50.5$_{\pm0.56}$ |
| | | 2 | 37.6$_{\pm0.76}$ | 27.5$_{\pm1.73}$ | 34.7$_{\pm1.14}$ | 36.4$_{\pm0.23}$ | 40.4$_{\pm0.30}$ | 41.8$_{\pm0.22}$ | 43.0$_{\pm0.92}$ | 43.4$_{\pm1.00}$ | 44.3$_{\pm0.29}$ | 44.7$_{\pm0.59}$ |
| | | 3 | 30.6$_{\pm0.62}$ | 21.0$_{\pm1.93}$ | 27.7$_{\pm1.14}$ | 29.6$_{\pm0.38}$ | 33.3$_{\pm0.36}$ | 34.3$_{\pm0.64}$ | 35.3$_{\pm0.60}$ | 35.0$_{\pm1.10}$ | 36.2$_{\pm0.29}$ | 36.6$_{\pm0.88}$ |
| | | 4 | 28.0$_{\pm0.69}$ | 18.9$_{\pm1.91}$ | 24.7$_{\pm0.56}$ | 27.0$_{\pm0.24}$ | 30.8$_{\pm0.39}$ | 32.0$_{\pm0.68}$ | 32.3$_{\pm0.70}$ | 32.3$_{\pm0.67}$ | 33.7$_{\pm0.70}$ | 34.6$_{\pm0.47}$ |
| | | 5 | 20.0$_{\pm1.16}$ | 13.7$_{\pm1.74}$ | 17.7$_{\pm0.41}$ | 20.1$_{\pm0.12}$ | 22.8$_{\pm0.13}$ | 24.0$_{\pm1.45}$ | 24.4$_{\pm0.56}$ | 24.5$_{\pm0.55}$ | 25.7$_{\pm1.09}$ | 26.4$_{\pm0.76}$ |
| | Frost | 1 | 37.5$_{\pm0.25}$ | 32.7$_{\pm0.24}$ | 37.9$_{\pm0.66}$ | 39.7$_{\pm0.26}$ | 41.9$_{\pm0.48}$ | 43.8$_{\pm0.58}$ | 44.3$_{\pm0.71}$ | 45.7$_{\pm0.60}$ | 46.3$_{\pm0.65}$ | 46.9$_{\pm0.51}$ |
| | | 2 | 33.1$_{\pm0.62}$ | 27.7$_{\pm0.46}$ | 33.1$_{\pm0.93}$ | 35.0$_{\pm0.59}$ | 37.6$_{\pm0.36}$ | 39.7$_{\pm0.62}$ | 39.6$_{\pm0.74}$ | 41.2$_{\pm0.24}$ | 42.0$_{\pm0.85}$ | 42.5$_{\pm0.34}$ |
| | | 3 | 29.6$_{\pm0.58}$ | 24.0$_{\pm0.80}$ | 29.2$_{\pm0.63}$ | 31.2$_{\pm0.56}$ | 33.3$_{\pm0.64}$ | 35.4$_{\pm0.43}$ | 35.9$_{\pm0.65}$ | 37.1$_{\pm0.35}$ | 38.2$_{\pm0.89}$ | 38.3$_{\pm0.33}$ |
| | | 4 | 30.5$_{\pm0.86}$ | 23.9$_{\pm0.54}$ | 29.6$_{\pm0.61}$ | 31.5$_{\pm0.45}$ | 34.1$_{\pm0.41}$ | 36.1$_{\pm0.51}$ | 36.3$_{\pm0.65}$ | 37.7$_{\pm0.15}$ | 38.8$_{\pm0.74}$ | 39.0$_{\pm0.75}$ |
| | | 5 | 27.9$_{\pm0.88}$ | 21.0$_{\pm0.71}$ | 26.7$_{\pm0.23}$ | 29.1$_{\pm0.73}$ | 31.6$_{\pm0.44}$ | 33.1$_{\pm0.51}$ | 33.3$_{\pm0.42}$ | 35.2$_{\pm0.40}$ | 35.6$_{\pm0.87}$ | 35.7$_{\pm0.88}$ |
| | Snow | 1 | 27.9$_{\pm0.58}$ | 26.0$_{\pm0.74}$ | 29.7$_{\pm1.44}$ | 31.8$_{\pm0.84}$ | 34.8$_{\pm0.41}$ | 35.2$_{\pm0.96}$ | 35.4$_{\pm1.16}$ | 37.3$_{\pm0.71}$ | 36.5$_{\pm0.94}$ | 37.5$_{\pm0.65}$ |
| | | 2 | 19.3$_{\pm0.49}$ | 18.4$_{\pm0.38}$ | 21.4$_{\pm1.28}$ | 22.8$_{\pm1.07}$ | 25.7$_{\pm0.12}$ | 25.8$_{\pm0.52}$ | 26.4$_{\pm0.85}$ | 27.9$_{\pm0.81}$ | 27.9$_{\pm0.96}$ | 28.2$_{\pm0.77}$ |
| | | 3 | 19.3$_{\pm0.36}$ | 17.5$_{\pm0.34}$ | 20.2$_{\pm1.12}$ | 22.6$_{\pm1.02}$ | 25.6$_{\pm0.54}$ | 25.6$_{\pm0.40}$ | 26.1$_{\pm0.64}$ | 27.9$_{\pm0.99}$ | 26.7$_{\pm0.98}$ | 26.8$_{\pm0.67}$ |
| | | 4 | 14.6$_{\pm0.50}$ | 13.7$_{\pm0.48}$ | 15.8$_{\pm1.13}$ | 18.0$_{\pm1.00}$ | 20.9$_{\pm0.81}$ | 20.6$_{\pm0.42}$ | 20.9$_{\pm0.81}$ | 22.6$_{\pm0.95}$ | 21.5$_{\pm1.08}$ | 21.8$_{\pm0.74}$ |
| | | 5 | 14.7$_{\pm0.56}$ | 13.5$_{\pm0.25}$ | 15.7$_{\pm1.45}$ | 17.4$_{\pm0.54}$ | 19.8$_{\pm0.88}$ | 20.1$_{\pm0.09}$ | 20.7$_{\pm0.76}$ | 22.0$_{\pm0.83}$ | 21.8$_{\pm1.22}$ | 22.0$_{\pm1.03}$ |