# OpenReview forum: "Foveated Dynamic Transformer: Robust and Efficient Perception Inspired by the Human Visual System"
_ICLR.cc/2025/Conference — ICLR 2025 Conference Withdrawn Submission_

### Official Review · Reviewer_Dnxr · 2024-10-30

**Soundness:** 2
**Presentation:** 3
**Contribution:** 2
**Rating:** 3
**Confidence:** 4

**Summary:**

Inspired by Human Visual System (HVS), the authors propose a modified transformer architecture. The proposed architecture enhances robustness against short learning, adversarial attacks and common corruptions compared to other transformer(s) (specifically DeiT), while reducing computational costs.

**Strengths:**

1. This paper introduces a novel transformer variant inspired by the Human Visual System (HVS), integrating multi-resolution processing and fixation selection to enhance robustness to adversarial attacks and common corruptions.
2. With a 34\% reduction in computational cost, the proposed model pushed the boundaries of state-of-the-art model architectures and shows potential for real-time image processing applications.
4. The paper’s figures and tables enhance clarity and strengthen the presentation.

**Weaknesses:**

1. For Introduction(and related work in appendix) section: The authors omit many relevant works, failing to convince reviewer they are up-to-date with state-of-the-art methods in this area. The lack of discussion and comparison with previous work may weaken the paper's impact.

2. For experimental section: This paper lacks a comparison with other HVS-inspired models. While it compares performance to DeiT, the authors do not explain why. Is it due to DeiT's similar parameter count or MACs?

Saccdic mechanism compared with FIX model:
* Liu, J., Bu, Y., Tso, D., & Qiu, Q. (2023). Improved Efficiency Based on Learned Saccade and Continuous Scene Reconstruction From Foveated Visual Sampling. ICLR.
* Schwinn, L., Precup, D., Eskofier, B., & Zanca, D. (2022). Behind the machine's gaze: Neural networks with biologically-inspired constraints exhibit human-like visual attention. arXiv preprint arXiv:2204.09093.
* Elsayed, G., Kornblith, S., & Le, Q. V. (2019). Saccader: Improving accuracy of hard attention models for vision. NeurIPS.

HVS inpired works improve Adverserial Robustness:
* Shah, M., Kashaf, A., & Raj, B. (2024). Training on foveated images improves robustness to adversarial attacks. NeurIPS.
* Wang & Cottrell (2017). Central and peripheral vision for scene recognition: A neurocomputational modeling exploration. Journal of Vision.
* Cheung, Weiss & Olshausen (2017). Emergence of foveal image sampling from learning to attend in visual scenes. ICLR.
* Gant, Banburski & Deza (2022). Evaluating the adversarial robustness of a foveated texture transform module in a CNN. SVRHM.
* Reddy, Banburski, Pant & Poggio (2020). Biologically inspired mechanisms for adversarial robustness. NeurIPS.
* Harrington & Deza (2022). Finding Biological Plausibility for Adversarially Robust Features via Metameric Tasks. ICLR.

**Questions:**

1. Could you clarify what $x^{i,j}$ represents in Equation (8)? It doesn’t seem to be annotated in Figure 2 or explained in the text.
2. Based on my understanding, FDT should select multiple fixation patches simultaneously, so I'm unsure how the ordered scanpath is derived in Figure 9. Was the scanpath manually connected between several hotspots in the heat map?
3. In section 2.2, I do not understand why author "outputs two logits for binary fixation decisions". Will one with threshold work?
4. This work seems more like a hard attention ViT than an HVS-based ViT. Typically, an HVS model would include foveal-peripheral vision and a saccadic mechanism. A multi-layer CNN may not be called foveal-peripheral (otherwise, any CNNs could claim to emulate the HVS), and the saccadic mechanism usually involves a sequence of actions tracking regions of interest.
5. In Table 1, while FDT shows slightly better performance than DeiT on common corruptions, I’m curious about the computational budget for FDT in this context.
6. I could be mistaken, but isn’t it ImageNet-C, rather than ImageNet-O, that includes common corruptions?

---

### Official Review · Reviewer_qMhb · 2024-11-03

**Soundness:** 2
**Presentation:** 1
**Contribution:** 2
**Rating:** 3
**Confidence:** 4

**Summary:**

The paper introduces the Foveated Dynamic Transformer (FDT), a vision transformer model inspired by the human visual system’s foveation and fixation mechanisms. FDT integrates two key modules: a foveation module that captures multi-scale information with high-resolution central and low-resolution peripheral features, and a fixation module that dynamically selects fixation points for further processing based on relevance. This structure reduces computation by 34% over DeiT, achieved through selective processing of informative regions. Other benefits of the bio-plausible mechanism related to adversarial robustness and robustness to image corruptions have been studied in the paper.

**Strengths:**

1. Single pass transformer based architecture that exhibits bio-inspired actions of saccades and foveation is a potential good direction, both as a bio-plausible way of learning and also as a method that can potentially address different problems related to robustness.
2. Sections 3.5 and 3.8.
3. The paper in the experimental section does touch upon a lot of relevant problems in DL, which is a good attempt. But for adversarial robustness and computational efficiency, detailed experiments will further back the claim.

**Weaknesses:**

1. The overall presentation of the method can be improved. The neuroscience principles of foveation and fixation are introduced, but their translation into the transformer blocks lacks depth, leaving the connection between biological mechanisms and deep learning components of a transformer underexplored. It would be good if in Section 2, a flow chart can be provided to show the general flow of logic and how the concepts from the HVS have been realized with the proposed architecture. Figure 1 is not mentioned in the text.
2. The claims for adversarial robustness and computational efficiency are not well supported. For computation efficiency please include concepts such as any time inference and budgeted classification as outlined in (a) and (b). 3.4 needs to be well sketched out since this is one of the key contributions of the paper. It will be good if a derivation be shown as to how FDT achieves a 34 % improvement upon DeiT. The derivation must track the results shown. For any one of the three variants should be fine. (tiny or small or base)
3. For adversarial robustness, I do not see transformer based attacks such as Token Gradient Regularization [CVPR 2023] and Towards transferable adversarial attacks on image and video transformers [IEEE Transactions on Image Processing] shown. Since these are transformer based attacks, they will serve as a better test for the proposed method over attacks that have specifically tailored for CNN based network architectures.
4. Many different attacks are chosen, no intuition is provided as to the rationale behind. Please define the threat model and explain the rationale behind the chosen set of adversarial attacks. Some heat maps if provided as predictions of Deit and FDT on different adversarial samples, will be helpful to analyze the effect of foveation and fixation modules. The heat map patterns will help us understand how the addition of these modules influence the predictions. For a clean sample, heat maps can be visualized. Now for the same sample, generate adversarial counterparts based of say PGD and some other attack. Regenerate the heat maps. See how the fixation points change. Please follow method outlined in [k].
5. Comparison to Adversarially trained Vision Transformers (i) and Robust Vision Transformer (f) are missing in terms of comparison. Since these one are Adversarially trained networks and the other has been proposed as a vision transformer with robust properties, these will serve as proper vetted baselines for comparisons.
6. (a) and (b) have showed their method is applicable for various well accepted CNN backbones for example ResNet, Efficient Net, etc. Too much focus on one architecture baseline (DeiT) is shown here. It would be good if the method can be extended to other vision transformer backbones such as a standard Vision Transformer itself. Why was a DeiT chosen as the baseline?
7. Comparing attention maps across focal modulation networks (c), focal transformers (d), Swin transformers (e), and FDT would be highly beneficial, as each of these models implements attention by selectively processing tokens in a distinct manner. Such a comparison could provide valuable insights into the model behavior and attention distribution unique to each approach, especially regarding efficiency and interpretability. Additionally, since FoveaTer has shown advantages in adversarial robustness, it would be helpful to include an intuition on how FDT differs from FoveaTer (j) to enhance its robustness. Explaining these differences more fully would clarify the specific contributions of FDT within the broader landscape of selective attention methods.
8. Missing citations and discussion of some very critical related work. Please incorporate these citations in related work and please feel free to refer to these to clarify the questions asked.

### References

### References

a. **Glance and Focus Networks for Dynamic Visual Recognition** – *T-PAMI 2022*

b. **Glance and Focus: A Dynamic Approach to Reducing Spatial Redundancy in Image Classification** – *NeurIPS 2020*

c. **Focal Modulation Networks** – *NeurIPS 2022*

d. **Focal Attention for Long-Range Interactions in Vision Transformers** – *NeurIPS 2021*

e. **Swin Transformer: Hierarchical Vision Transformer using Shifted Window** – *arXiv preprint*

f. **Towards Robust Vision Transformer** – *CVPR 2022*

g. **Evading Defenses to Transferable Adversarial Examples by Translation-Invariant Attacks** – *CVPR 2018*

h. **Enhancing the Transferability of Adversarial Attacks through Variance Tuning** – *CVPR 2021*

i. **When Adversarial Training Meets Vision Transformers: Recipes from Training to Architecture** – *NeurIPS 2022*

j. **FoveaTer: Foveated Transformer for Image Classification** – *arXiv preprint*

k. **On Inherent Adversarial Robustness of Active Vision Systems** – *arXiv preprint*

**Questions:**

1. Please clarify the points mentioned in the Weakness section.
2. In the abstract, why is metabolic energy mentioned? What is the significance?
3. Results in Section 3.1 is slightly misleading. The improvements are presented in terms of percentage. The real numbers are presented in the Appendix. Overall which library was used to evaluate the robustness of the method. Also why are CW numbers so much better than PGD?  As in as a white box attack CW and PGD should be very close, bringing the accuracy down close to 0, at least for DeiT.
4. Please try attacks (g) and (h) as mentioned in the Weakness section. These are stronger attacks than MIGFSM.
5. A major contradiction that I notice is that for humans, we first scan the environment (saccades) and once we figure out the key regions, we foveate onto those to truly understand the content. This network seems to do the opposite, first foveate and then fixate. Please explain.
6. Is the fixation module referred to as the Dynamic Network? Is it mentioned anywhere?
7. Does the combination module have negative effect on the fixated regions? Since at this stage both fixated and non-fixated tokens are processed together for dimension matching. Does this not defeat the purpose of the previous modules?
8. Why are results shown on ImageNet100 and not ImageNet-1K, which is a more established benchmark for the targeted task?

---

### Official Review · Reviewer_eqWe · 2024-11-03

**Soundness:** 2
**Presentation:** 1
**Contribution:** 2
**Rating:** 3
**Confidence:** 4

**Summary:**

This paper introduces the Foveated Dynamic Transformer (FDT), an architecture inspired by the human visual system (HVS). FDT aims to emulate the foveation and fixation mechanisms of the HVS using a single-pass strategy, facilitated by dedicated fixation and foveation modules. The foveation module encodes multi-scale information, while the fixation module filters out irrelevant information, optimizing the attention process. The authors hypothesize that FDT is robust against adversarial attacks, with experimental results supporting this claim. By adjusting the threshold (or budget) for fixation points, the FDT framework achieves computational efficiency gains with minimal performance loss.

**Strengths:**

This paper addresses a significant topic in the computer vision community by drawing inspiration from neuroscience. Specifically:
- It presents a novel framework addressing two critical aspects of computer vision systems: computational efficiency and adversarial robustness.
- The FDT framework incorporates concepts from human vision, potentially helping to overcome current limitations faced by non-neuro-inspired frameworks, such as convolutional neural networks.

**Weaknesses:**

However, the paper has several weaknesses that should be addressed before it is suitable for publication at a venue like ICLR. Specifically:

- **Structure and Clarity:** The paper is poorly organized and challenging to follow.
   - **Figures:** Figures should be referenced appropriately in the main text to enhance clarity. For instance, Figure 1 is not referenced at all, and Figure 2, which contains substantial information, could be better integrated into the main text to explain the framework's operation, rather than relying solely on its caption. The authors may benefit from approaching the text from the perspective of readers unfamiliar with the work, guiding them to figures as needed.
   - **Abbreviations:** Abbreviations like MHSA and GMACs are used repeatedly before they are fully explained. It’s essential to consider that readers may be new to the field, and all abbreviations and specialized terms should be clearly defined.
   - **Missing Details:** The paper lacks key details, such as the threat model used in the adversarial attack section.

- **Experimental Clarity:**
   - Several aspects of the experimental setup lack sufficient explanation. For instance, in Table 1, different security levels are referenced without a clear explanation of their meaning. While these terms may be familiar to experts in the exact field, they are new to many readers.
   - **Comparisons:** Many recent computer vision works have drawn on neuroscience inspiration, especially in modifying or leveraging CNNs. It is crucial to compare the proposed work to these studies in terms of accuracy, robustness, training and inference computation, and memory requirements. I have listed a few references below. The authors could look into venues such as the *GAZE meets ML workshop* at NeurIPS, which focuses on neuro-inspired computer vision techniques for more up to date references. It would also be beneficial to include specific studies like [5,6], which address the adversarial robustness of similar frameworks.

**Relevant References**

*[1] Jindal, S., Yadav, M., & Manduchi, R. "Spatio-Temporal Attention and Gaussian Processes for Personalized Video Gaze Estimation." IEEE/CVF CVPR, 2024.*

*[2] Gupta, A., et al. "Exploring the Zero-Shot Capabilities of Vision-Language Models for Improving Gaze Following." IEEE/CVF CVPR, 2024.*

*[3] Ibrayev, T., et al. "Exploring Foveation and Saccade for Improved Weakly-Supervised Localization." Gaze Meets ML Workshop, PMLR, 2024.*

*[4] Wang, Y., et al. "Glance and Focus: A Dynamic Approach to Reducing Spatial Redundancy in Image Classification." NeurIPS, 2020.
Luo, Y., et al. "Foveation-Based Mechanisms Alleviate Adversarial Examples." arXiv preprint, 2015.*

*[5] Mukherjee, A., Ibrayev, T., & Roy, K. "On Inherent Adversarial Robustness of Active Vision Systems." arXiv preprint, 2024.*

*[6] Luo, Y., et al. "Foveation-Based Mechanisms Alleviate Adversarial Examples." arXiv preprint, 2015*

**Questions:**

In addition to the weaknesses highlighted, here are questions that, if addressed, could significantly improve the reader’s understanding of the paper:

1. How does this framework compare to other frameworks that incorporate multi-resolution information, such as pyramidal networks?

2. How does the framework implement foveation and saccadic (fixation) movement? Shouldn’t fixations precede foveation to more accurately reflect human vision? Further, how does the proposed foveation module focus on specific areas?

3. Is feature inversion necessary within this framework? While it aids in explainability, it may not be required for model prediction at inference.

4. How were samples classified as easy, medium, or hard? Were standard definitions or existing classification methods, such as memorization studies, used?

5. The response time analysis claims a comparison to HVS response times in Table 4, but no such comparison is presented in this table.

6. If the system is single-pass, how is the order of fixation points determined?

7. Transformers typically require substantial training data or pre-trained models. Was a pre-trained model used here, given the relatively small training dataset?

8. How does patch size affect performance?

9. What is the significance of feature inversion in Figure 7?

---

### Note · Authors · 2024-11-22

I have read and agree with the venue's withdrawal policy on behalf of myself and my co-authors.